

# Response of soil respiration and soil microbial biomass carbon and nitrogen to grazing management in the Loess Plateau, China

Zhen Wang[1,2], Xiuli Wan[1], Mei Tian[1], Xiaoyan Wang[1], Junbo Chen[1], Xianjiang Chen[1],
Shenghua Chang[1], Fujiang Hou[1,2]

[1]State Key Laboratory of Grassland Agro-Ecosystems; College of Pastoral Agriculture Science and
Technology, Lanzhou University, Lanzhou 730020, Gansu, China

[2]Key Laboratory of Grassland Agro-Ecosystem, Ministry of Agriculture; Key Laboratory of Grassland

Livestock Industry Innovation, Ministry of Agriculture; College of Pastoral Agriculture Science and
Technology, Lanzhou University, Lanzhou 730020, Gansu, China

*Correspondence to*: Fujiang Hou (cyhoufj@lzu.edu.cn)

## Abstract

Grassland covers more than a third of the earth's terrestrial surface. Grazing management can affect
grassland carbon dynamics and soil microbial biomass, yet limited information is available on the
effects of grassland management on carbon dioxide efflux and soil microbial biomass carbon (SMBC)
and nitrogen (SMBN). During 2010 and 2011, soil respiration (Rs), SMBC, and SMBN, as well as
different abiotic and biotic factors were measured after long term rotational grazing (nine years) on the

grasslands of the semi-arid Loess Plateau, China. Grazing management included different grazing
intensities and seasonal grazing patterns (in summer or winter). Stocking rates of 0, 2.7, 5.6, and 8.7
sheep ha$^{-1}$ were used as grazing intensities, and warm-season grazing and cold-season grazing by sheep
during summer and winter from 2010 to 2011 were used as grazing patterns. We hypothesized that the
different seasonal grazing patterns and grazing intensities would affect Rs in a semi-arid grassland

ecosystem. Our results indicated that grazing management significantly affected the rate of Rs, which
supports our hypothesis. Grazing intensities tended to increase SMBC, but had no effect on SMBN. We
also found that SMBC in cold season grazing plots was higher than that in the warm season grazing
plots. However, variation in grazing patterns had little effect on SMBN. Furthermore, a structural
equation model indicated that the aboveground biomass and soil microbial biomass were two important





biotic factors that controlled Rs. Soil temperature (ST) and soil moisture (SM), which were affected by grazing intensity and patterns, were significant abiotic factors affecting Rs and soil microbial biomass. Our observations suggest that grazing management may change soil carbon sequestration rates in grassland ecosystems, because of changes in the aboveground plant and soil microbial biomass.

## 1. Introduction

Grassland covers more than a third of the earth's terrestrial surface, and traditionally supports domestic livestock grazing in extensive agriculture grazing systems (Morgan et al. 2007). Grasslands can provide important ecosystem services to humans, because of their large area and excellent ability to sequester

and store carbon (C) (Zhao et al. 2017). China's grasslands cover 6 – 8% of the total grassland area of the world, and make a significant contribution to the world's carbon storage, thereby significantly affecting global carbon cycles (Ni 2002). Soil respiration (Rs) is the second largest carbon (C) flux between the atmosphere and terrestrial biomes (Cox et al. 2000; Wan et al. 2007) and consists mainly of microbial and root respiration (Davidson and Janssens 2006; Jia et al. 2007; Ru et al. 2017).

Numerous studies have indicated that biological and environmental variables, such as precipitation variability (Xu et al. 2016; Hawkes et al. 2017), soil temperature (Thomey et al. 2011), soil water content (Hawkes, et al. 2017), climate, and microbial community composition (Monson et al. 2006) are the main factors that determine soil and ecosystem respiration in grassland ecosystems. In addition, human activities are also strongly modifying biogeochemical cycles in many ways (Wang and Fang

2009). For instance, the structure or species composition of plant communities, soil microclimate, soil chemical and physical properties, and global climate are changed by human activities, and these variables may all effect Rs rates (Raich and Schlesinger 1992).

Grazing is common in natural grasslands, and is often associated with traditional ways of life (McSherry and Ritchie 2013; Wang et al. 2017). Almost all grasslands in East Asia are subject to some

degree of grazing pressure, and grazing intensity alters not only the Rs rate, but also the temperature dependence of soil $CO_2$ efflux (Cao et al. 2004). Although a growing number of studies show that the health of grassland ecosystems strongly depends on grassland management strategies, such as grazing and grazing exclusion (Chen et al. 2015; Deng et al. 2017), the C dynamics in grassland ecosystems with different grazing management strategies have not been well characterized (McSherry and Ritchie

2013; Deng et al. 2017). Indeed, previous studies have shown that increased stocking rates did not





contribute to elevated $CO_2$ efflux within native vegetation pastures (Liebig et al. 2013), both no grazing and heavy grazing significantly decreased C fixation of the steppe grassland (Rong et al. 2017), and Rs rate was enhanced significantly in fenced plots (Jia et al. 2007). Therefore, there is still controversy regarding how grazing management affects Rs. Moreover, warm (summer) and cold (winter) season

grazing are the two types of grazing patterns in place in the Loess Plateau grasslands (Chen et al. 2015). Chen *et al.* (2015) found that warm season grazing reduced Rs, whereas Wang et al. (2017) believe cold-season grazing significantly increases Rs. Hence, limited information is available on the effects of grazing pattern on soil respiration, and on how Rs responds to the grazing-mediated changes in plant properties and soil conditions, particularly in the natural grasslands of the Loess Plateau. Grasslands are

widely distributed in the Loess Plateau, accounting for approximately 40% of the total area (Wang et al. 2017). Drought, soil erosion, and desertification are ongoing concerns for local inhabitants whose agricultural activities and associated income rely largely on the availability of precipitation (Fan et al. 2015). The Loess Plateau is characterized by continuous and widespread stress, including over-grazing (Fu et al. 2000). Several studies on Rs have been carried out in the different ecosystems of the Loess

Plateau, including cropland ecosystems (Li et al. 2010; Zhang et al. 2011), forest ecosystems (Yan et al. 2013; Shi et al. 2014), and the transitional zone between forest and grassland (Shi et al. 2011). Hence, quantifying the effects of grazing intensity and grazing seasons on total soil respiration is critical to accurately estimate the carbon balance of grassland ecosystems, and to better understand how global changes affect grazing in the Loess Plateau.

As an important component of terrestrial ecosystems, soil microorganisms play an important role in global nutrient cycling and organic matter decomposition (Wang et al. 2013). Furthermore, soil microbial biomass carbon (SMBC) adjusts the balance between the release of soil respiration and its sequestration in soil organic matter in terrestrial ecosystems (Lange et al. 2015; Thakur *et al.* 2015). In addition, microbial biomass can be used to assess soil quality of different types of vegetation. Soil N is

a key factor in the regulation of soil C sequestration (Deng et al. 2017), and increased $CO_2$ has been found to reduce available soil nitrogen, exacerbate nitrogen constraints on microbes, and reduce microbial respiration per unit biomass (Hu et al. 2001). Gallardo and Schlesinger (1992) found a succession in the control of microbial biomass from nitrogen to carbon when the ratio of carbon to nitrogen decreased during desertification. Fu et al. (2012) found that grazing significantly decreased

SMBC and soil microbial biomass nitrogen (SMBN) in an alpine meadow, while Wang et al. (2006)

found that cattle grazing increased microbial biomass. Limited evaluations of soil microbial biomass in semi-arid grasslands, along with divergent outcomes of grazing, underscore the need for additional research, especially on grazing intensity (GI) and grazing patterns (GP). Therefore, studies on linkages between soil microbial biomass and environmental parameters would provide a better understanding of

the factors that control nutrient cycling in grassland ecosystems.

Given this context, we hypothesized that grazing management (grazing intensity and seasonal grazing pattern) could significantly change Rs and soil microbial biomass carbon and nitrogen by disturbing biotic factors (above-ground and belowground biomass) and abiotic factors (soil temperature and soil moisture) in a semi-arid grassland ecosystem. To test this, a two-year study based on a long

term rotational grazing experiment was conducted. We aimed to assess Rs from grassland soils under different grazing intensities and two grazing patterns, and analyzed the effects of these grazing intensities and patterns on soil microbial biomass carbon and soil microbial biomass nitrogen.

## 2. Materials and method

### 2.1 Site description

The study site is located in Huanxian Grassland Ecosystem Trial Station in eastern Gansu province, northwest China (37.14 °N, 106.84 °E, 1650m a.s.l.), which is the core region of the Loess Plateau (Fig. 1). The average annual rainfall is 360 mm and >70% of rainfall takes place from middle June through September (Fig. 2). The average annual air temperature of 7.1 ℃ with maximum temperature in July

and minimum temperature in January. Average annual potential evaporation is 1993 mm. Spring and autumn in the study area are typically short; summer is hot and humid; and winter is long and cold. The soil is classified as sandy, free-draining loess and the rangeland is a typical temperate steppe (Hou et al 2002). The dominant species are *Stipa bungeana, Lespedeza davurica, Pennisetumflaccidum, and Artemisia capillaries , and Setariavirdis* (Hou et al., 2002). Annual mean air temperatures were 8.7 and

7.9 ℃ in 2010 and 2011, respectively (Fig. 2). Mean monthly precipitation was 23.2 and 27.8 mm in 2010 and 2011 (Fig. 2), respectively. Precipitation was much higher in September than in other months in 2011.

### 2.2 Experimental design

Two areas which had similar topographic conditions, vegetation composition and cover were

selected for warm season (summer) grazing and cold season (winter) grazing, respectively. Each of

them was divided into twelve 0.5 ha enclosed experiment plots for a rotational grazing trial of Tan sheep,. Four stocking rates of 0, 2.7, 5.3, 8.7 sheep ha$^{-1}$ completely randomly arranged in experiment plots, 3 replicates, respectively. Warm season grazing plots were rotationally grazed from June to September. One cycle of rotational grazing is 30 days with 10 days for grazing and 20 days for rest,

three rotations for total 90 days. Cold season grazing started from mid-November to late December for 48 days. One cycle of rotational grazing is 24 days with 8 days for grazing and 16 days for rest, two rotations for total 48 days. The experiment of the rotational grazing system started from 2001.

### 2.3 Rs measurement

In 2010 and 2011, after 9 years rotational grazing, soil efflux chamber was attached to a gas analyzer (LI-COR 6400, Lincoln, NE, USA) was used to measure Rs in our experiment plots. There were three sampling transects (50 m long) 30-40 m apart were randomly established in each experiment plots. Two PVC collars (11.0 cm diameter, 5.0 cm height) were randomly inserted into the soil along each sampling transects. For more information about Rs measurement and PVC collars installation see Chen

*et al* (2015). Before the measurement, all aboveground vegetation in each PVC collar were clipped to ensure only Rs were measured. Rs were continuously measured for 6 days during the middle of May, September, and December, from 2010 to 2011 in the grazing experiment plots, respectively. All daily change of Rs were measured 2-hours intervals from 6:00 am to 22:00 pm.

### 2.4 Soil temperature and soil moisture

Soil temperature (ST) at 10 cm depth was measured using a thermocouple probe which attached to the gas analyzer adjacent to each PVC collar during the time of Rs measurement. At the nearby PVC collar, soil moisture (SM) at the top 10.0 cm was measured by the gravimetric method, which involves drying soil samples at 105 ℃ for 48 h. Soil samples used to measure soil moisture was collected concurrently within days of Rs sampling .

### 2.5 Soil sampling and biomass measurements

Soil samples were collected at random locations adjacent to each Rs measurement site at 5 cm and 10 cm depth during the Rs determination period. After soil samples were coarsely sieved (4.75 mm) to remove rocks and large roots, it were sealed in plastic bags and brought to the laboratory for analysis as soon as possible. Aboveground biomass (AGB) was estimated by cutting all vegetation in 1m×1 m

quadrats (six per plot) after the plots were grazed during the second rotation in early September in 2010



and 2011, when the aboveground biomass reached peaks. Samples were oven dried at 65 ℃ until a

constant weight was obtained. Once AGB and litter were harvested, soil cores (10 cm depth, 10.0 cm

diameter) to a depth of 1 m were collected by a soil auger to calculate belowground biomass (BGB) in

each quadrats. For more details about BGB evaluation see *Chen et al.* (2015).

**2.6 Soil microbial C and N**

Soil microbial biomass carbon (SMBC) and nitrogen (SMBN) were determined using a chloroform

fumigation-extraction procedure, and were calculated using the difference in dissolved organic carbon

(DOC) and dissolved organic nitrogen (DON) between fumigated and non-fumigated soil subsamples

(Brookes et al. 1985; Vance et al. 1987). Briefly, 10 g soil samples were fumigated with chloroform for

24 h in a vacuum desiccator, and another 10 g served as non-fumigated controls. C and N were

extracted with 50 ml of 0.5 M $K_2SO_4$ for 30 min, from fumigated and non-fumigated samples, and the

extracts were filtered and frozen at -20 ℃ before analysis with a Total Dissolved Organic Carbon and

Nitrogen Analyzer-multi NC 2100S (Analytik Jena AG, Analytik JenaCo., Jena, Germany).

**2.7 Statistical analysis**

One-way ANOVA analyses followed by a multi-comparison of a least significant difference (LSD)

test was performed to examine the effects of grazing intensity (0, 2.7, 5.3, and 8.7 sheep $ha^{-1}$), and

grazing patterns (warm and cold season grazing) on ST, SM, and daily mean variation in Rs. One-way

ANOVA analyses with LSD tests were also performed to examine the effect of sampling time

(May/September/December) and year (2010/2011) on ST ( ℃), SM (%), Rs ($\mu mol$ $CO_2$ $m^{-2}$ $s^{-1}$), SMBC

(g $kg^{-1}$), and SMBN (g $kg^{-1}$). General linear models (GLMs) were used to assess the effects of grazing

intensity (GI, 0, 2.7, 5.3, and 8.7 sheep $ha^{-1}$), grazing pattern (GP, warm and cold season grazing),

sampling time (May/September/December), year (2010/2011), and their interactions on Rs, SMBC, and

SMBN. In these models, GI, GP,sampling time, and year were treated as fixed factors, and block was

treated as a random factor. Significant differences for all statistical tests were evaluated at the level of P

≤ 0.05. To examine the temperature sensitivity of Rs, we conducted regression analyses using Rs = $ae^{bT}$,

where Rs is soil respiration, T is soil temperature, coefficient a is the intercept of soil respiration when

the temperature is 0 ℃, and coefficient b represents the temperature sensitivity of soil respiration,

which was used to calculate a respiration quotient $Q_{10} = e^{10b}$ (Luo et al. 2001). Two-sample *t*-test for

the means was used to identify the significance of the difference between $Q_{10}$ values under different

grazing patterns. Unless specified, the significance level was set at P < 0.05 and uncertainty (±) always





refers to a 95% confidence interval. All statistical analyses were conducted using SPSS 17.0 (SPSS Inc.,

Chicago, IL, USA). Structural equation modeling (SEM) was used to explore the pathways of how GI

and GP influence biotic and abiotic factors, and how this affected Rs and soil microbial biomass carbon

and nitrogen. We first considered a full model that included all possible pathways, and then

5       sequentially eliminated non-significant pathways until we attained the final model. We used $\chi^2$ test,

Akaike information criteria (AIC), and the root mean square error of approximation to evaluate the fit

of model. SEM analyses were conducted using AMOS 17.0 (SPSS Inc., Chicago, IL, USA).

### 3. Results

### 3.1 Daily and seasonal dynamics of Rs rate under four different grazing intensities and two
### grazing patterns

We measured the diurnal courses of hourly mean $CO_2$ exchange flux in warm season grazing and

cold season grazing plots (under grazing intensities of 0, 2.7, 5.3, and 8.7 sheep ha$^{-1}$) on clear days. The

daily maximum, minimum, and total Rs rates under the four grazing intensities in the two grazing

seasons from May 2010 to December 2011 are shown in Table S1 and Table S2, respectively. There

were no significant differences in daily Rs rates under the different grazing intensities in the warm

season grazing plots (P > 0.05). However, Rs rates in cold season grazing plots with 0 sheep ha$^{-1}$

(control plots) were significantly lower (45%) compared to other grazing treatments in 2010 (Table S2,

P < 0.05), and significantly higher (26%) in September 2011 (Table S2, P < 0.05). To test if grazing

pattern (warm or cold season grazing) induced significant differences in daily Rs rates, the

corresponding data from the warm grazing season and cold grazing season plots were compared with

one-way ANOVA. There were significant differences in daily Rs rates between warm grazing season

and cold grazing season plots (P < 0.001, data not shown).

We further analyzed responses of monthly mean Rs rates for each year in warm grazing season and

cold grazing season plots (Fig. 3 and 4). In warm season grazing plots, the seasonal dynamic of Rs

rates with grazing intensities of 2.7, 5.3, and 8.7 sheep ha$^{-1}$ were similar to those in control plots (0

sheep/ha$^{-1}$), and mean Rs rates were the highest (P < 0.05) in September 2011. Compared with control

plots (0 sheep ha$^{-1}$), seasonal dynamics of Rs with grazing intensities of 2.7, 5.3, 8.7 sheep ha$^{-1}$ were

similar in cold season grazing plots, and the maximum Rs value was approximately 1.6 μmol $CO_2$ m$^{-2}$

s$^{-1}$ in May in 2010 and 2011. When comparing grazing patterns in 2010 and 2011, Rs in cold season





grazing plots was markedly higher than that in warm season grazing plots (P < 0.001). GI, GP, year, sampling time, and their interactions significantly affected Rs (Table 1, P < 0.05).

### 3.2 SMBC and SMBN

SMBC and SMBN were both characterized by pronounced temporal dynamics between and within seasons over the investigated time across the two years of 2010 and 2011 (Fig. 5 and 6). SMBC in 2010 was clearly the highest of all the experimental periods, both in warm season grazing and cold season grazing plots (Fig. 5, P < 0.001). The value of SMBC in May was significantly higher than that in September and December (Fig. 5, P = 0.002). SMBC in treatments with 2.7, 5.3, 8.7 sheep ha$^{-1}$ was

approximately 26% higher than in plots with 0 sheep ha$^{-1}$ (control plots) (Fig. 5, P = 0.019). Moreover, the concentration of SMBC in warm season grazing plots was 15.6% lower than in cold season grazing plots (Fig. 5, P = 0.022). The maximum value of SMBN content at the 0 - 5 cm soil depth was recorded in September 2011, in both warm season grazing and cold season grazing plots. However, the maximum value of SMBN content at the 5 - 10 cm soil depth was recorded in May 2010, in both warm

season grazing and cold season grazing plots. General linear models showed that GP and GI significantly affected SMBC (Table 1, P < 0.01), but there was no significant effect of the interaction between GP and GI on SMBC (Table 1, P = 0.28). SMBC was also affected by sampling time and year, and their interactions. There were also significant effects of interactions between GP and sampling time (Table 1, P < 0.001) and GP and year (Table 1, P = 0.002) on SMBC. SMBN was not affected by GP

(Table 1, P = 0.057) and grazing intensity (Table 1, P = 0.375), nor by the interaction between grazing pattern and grazing intensity (Table 1, P = 0.134). Sampling time (P <0.001, Table 1), year (P < 0.001, Table 1) and their interaction (P = 0.002, Table 1) significantly affected SMBN. There was no significant interaction effect of GP, GI, and sampling time or year on SMBC (P > 0.05, Table 1).

*Effects of grazing management and biotic factors on the soil microclimate and Rs*

The diel variation in soil temperature (ST), measured at 2 h intervals at depths of 0 - 5 cm and 5 - 10 cm, changed significantly over time (Fig. S1, P < 0.05). There was no significant difference in ST in plots with different grazing intensities (2.7, 5.3, 8.7 sheep ha$^{-1}$) compared to the control plots (0 sheep ha$^{-1}$) (P = 0.63). The ST varied significantly (P < 0.001) by year, with values the highest in 2011 and the lowest in 2010 (Fig. S1). With respect to grazing pattern (warm or cold season grazing), ST in the cold





season plots was 14.67% higher (P = 0.02) than that in the warm season plots (Fig. S1). The annual

mean SM in 2011 was 7.90% higher (P < 0.001) than in 2010 (Fig. S2). SM in September was higher

than in other months (Fig. S2, P < 0.001). Although grazing intensities did not affect soil moisture (P =

0.87), plots with a grazing intensity of 8.7 sheep ha$^{-1}$ had higher SM (17.5%) than control plots (0

5     sheep ha$^{-1}$). There were no effects of grazing pattern (P = 0.11) on SM. The GI treatments resulted in

significantly lower mean AGB (P = 0.011), but did not affect BGB (P = 0.279). Across the two years of

2010 and 2011, mean AGB decreased by 23.78% (P = 0.002), but mean BGB increased by 33.14% (P <

0.001). Although there were no significant effects of grazing patterns on AGB (P = 0.051) and BGB (P

= 0.059), both AGB and BGB were higher in cold season grazing plots than in warm season grazing

plots (by 13.47% and 20.11%, respectively).

During the whole experimental period, a significant exponential positive correlation between the

rates of Rs change and ST was found in both warm season grazing plots and cold season grazing plots,

indicating that Rs is strongly coupled with ST (P < 0.001, Table 2). There was a clear difference in the

$Q_{10}$ values of Rs among the four grazing treatments both in warm and cold grazing season plots from

2010 to 2011 (P < 0.001, Table 2). With regard to the grazing patterns, $Q_{10}$ values of warm grazing

season grazing plots were significantly higher than those of cold season grazing plots (Table 2).

Structural equation models (SEMs) showed that the variance of Rs was directly and indirectly

explained by GP and/or GI, which, together, could explain 57% of the total variation of Rs in the

grazing ecosystem (Fig. 7 and 8). GI significantly affected Rs due to a decrease in above ground

biomass (AGB) and belowground biomass (BGB), which had a direct positive and negative effect on

Rs, respectively (Fig. 7). In addition, AGB positively influenced Rs through increased SMBC. GI also

increased soil respiration by directly affecting SMBC. According to the SEMs, the changes in Rs were

directly affected by changes in grazing patterns, and by ST, SM, SMBC and SMBN, which were

affected by GP. Moreover, ST and SM could directly or indirectly affect Rs through changes in SMBC

and SMBN (Fig. 7 and 8).

### 4.  DISSCUSSION

#### 4.1 Effect of grazing management on Rs

Our experimental results indicate that both warm-season grazing and cold-season grazing affect Rs,

which supports our hypothesis, namely that the different grazing patterns could affect Rs in a semi-arid



grassland ecosystem. Daily changes of Rs under different grazing patterns (warm and cold season) with four grazing intensities (0, 2.7, 5.3, 8.7 sheep ha$^{-1}$) in our study were in the range of daily Rs reported by several previous studies (Zhang et al. 2014; Wang et al. 2015; Rong et al. 2017). Our research revealed that the daily variation in Rs in cold season grazing plots was significantly higher (by 22.7%)

than that in warm season grazing plots, which explained more than 40% of the variation in daily Rs for all grazing treatments. This is consistent with the results obtained from a meta-analysis of Tibetan grasslands (Wang et al. 2017).

We found that GI had no effect on daily variety of Rs in warm season grazing plots. However, it had a clear effect on cold season grazing plots. Since diel changes of soil respiration followed a unimodal

pattern through time, consistent with ST, this could be due to differences in sensitivity to temperature in grazing seasons, or could be due to spatial heterogeneity of Rs (Wang et al. 2013; Wang et al. 2017). In our study, daily trends in Rs were similar to previous reports from the same study site (Chen et al. 2015). Grazing modifies the grassland microclimate by removing plant material and compacting the soil, which leads to a warmer and drier microclimate (Rong et al. 2017). Our results also suggest that

GI markedly affected the AGB in the semi-arid grassland ecosystem, but did not affect BGB. This may be attributed to the fact that grazing reduces standing dead material, allowing more light to shine on the ground, thus favoring plant growth in the next year (Coughenour 1991; Wang et al. 2017). ST and water availability can directly affect soil respiration by altering activities of plant roots and soil microbes, and can indirectly affect soil respiration by changing plant growth and substrate supply (Wan

et al. 2007). In the context of this study, grazing patterns could significantly change ST and SM, which led to a significant increase in seasonal changes of Rs, which is in agreement with previous studies (Cui et al. 2014; Wang et al. 2017.) Several studies on the temperature sensitivity of Rs have been conducted in various ecosystems, both globally and across China (Luo et al. 2001; Curiel et al. 2004; Davidson et al. 2006; Chen et al. 2015; Chen et al. 2016). The value of $Q_{10}$ was 1.31 to 1.57 and 1.21 to 1.36 in

warm season grazing and cold season grazing plots with different grazing intensities, respectively, which is in close agreement with the results of Chen et al. (2015), who quantified the temperature sensitivity of nighttime $CO_2$ flux using the eddy covariance method in the same semi-arid grassland on the Loess Plateau. Although $Q_{10}$ in warm season grazing plots was higher than in cold season grazing plots, it is possible that the low soil temperatures and frozen soil water during the non-growing season

inhibited soil microbial activity, thus reducing $Q_{10}$ (Mahecha et al, 2010; Chen et al. 2016). The Q10



values on the global ecosystem scale vary, but had a median value of 1.4 (Mahecha et al. 2010). Our result is similar to this estimated median value at the global scale, indicating that future warming of the climate or future grazing management will exert great impacts on the respiration of semi-arid grassland ecosystems.

We can assume that differences in Rs between warm season grazing plots and cold season grazing plots areas are mainly due to the biological activity of plant roots, and abiotic factors such as ST and SM, which were affected by grazing managements. These results are supported by the SEM analysis in our study and previous research (Jiang et al. 2010; Wang et al. 2013; Chen et al. 2015; Xu et al. 2016). In addition, the effect of ST on Rs can be explained by the distribution of inter-annual precipitation. Ru

et al. (2017) found that the amount of precipitation in the middle growing season is more important than that in the early, late, or whole growing seasons in terms of regulating soil C release in grasslands. This result is confirmed in our study, i.e., maximum precipitation and maximum soil respiration rate occurred in September 2010 and 2011 in semi-arid grassland. Furthermore, grazing had comparatively stronger effects on potential microbial respiration than on standing biomass (Bagchi et al. 2017). Cattle

grazing increases microbial biomass and alters soil nematode communities in subtropical pastures (Wang, et al. 2006). Since soil respiration is a process of transferring organic C into inorganic C, the rate of Rs is ultimately controlled by the supply of C substrate (Xu et al. 2016; Bagchi et al. 2017). According to the SEM analysis, the Rs was strongly regulated by SMBC and SMBN, which was directly or indirectly controlled by grazing intensities and grazing patterns.

Overall, ST and SM were the main abiotic factors affecting Rs, while aboveground biomass and soil microbial biomass (which were affected by grazing management) were the main biotic factors affecting Rs in this grassland ecosystem. Distinguishing between the different and interactive impacts of the above factors on Rs will not only improve our understanding of the dynamics and patterns of soil respiration in terrestrial ecosystems, but will also facilitate projections of the responses of soil

respiration under global change.

**4.2 Effects of grazing management on soil microbial biomass**

      Previous studies indicate that the effects of grazing on soil microbial community sizes are largely

dependent on grazing intensity via both biotic and abiotic factors (Zhao et al. 2017). On one hand, our



study revealed that SMBC was higher at the beginning of the growing season (May) and in the middle

of the growing season (September) than in the dormant time (December), which is consistent with

previous studies conducted in the grassland ecosystem of the Trans-Himalaya (Bagchi et al. 2017), and

the Tibetan Plateau (Fu et al. 2012), which indicates that grazing may affect potential microbial activity,

thus affecting the stability of the soil carbon pool. Our analysis indicates that GI tended to increase

SMBC, while GI had no effect on SMBN. These results are in line with previous studies on wet

grassland grazing, which demonstrated a clear effect of grazing on increasing microbial biomass C

(Wang et al. 2006). Our findings imply that the soil microbial biomass possesses a stronger capacity for

high metabolic activity under GI compared with the soil microbial biomass under enclosed plots,

indicating that GI may select for more efficient enzymes to catalyze reactions than enclosed grazing

(Stark et al. 2015). The large increases in microbial biomass nutrients in grazed plots indicate that

grazing increased C availability to the soil microbial community, either directly through increased

inputs of waste products and litter, or indirectly through altering C allocation or flows in the

rhizosphere (Wang et al. 2006). Furthermore, our study also found that SMBC in cold season grazing

plots was significantly higher than in warm season grazing plots. This phenomenon might be explained

by the fact that cold-season grazing increased the soil temperature, which increased microbial biomass

(Lu et al. 2013; Wang et al. 2017).

Contrarily, it is interesting to ask to what extent the grazing management effects on microbial

biomass were connected with the responses of the soil microbial biomass to ST, SM, AGB, and BGB. It

is more likely that climatic differences between years significantly altered grazing responses

(Coughenour 1991). Rainfall was higher in September than in May and December, and soil temperature

and soil moisture varied significantly, with values the highest in 2011 and the lowest in 2010 in this

study. More importantly, the results of SEMs confirmed that different grazing seasons positively

affected ST and SM, both of which enhanced the soil microbial biomass. A grazing intensity of 8.7

sheep ha$^{-1}$ significantly increased soil moisture by 17.5% when compared with a grazing intensity of 0

sheep ha$^{-1}$ (Fig. S2). Storage of soil carbon is governed by the metabolic activity of soil microbes,

which is mediated by plant diversity via higher root inputs and other, as yet unidentified, mechanisms

(Lange et al. 2015). Carbon uptake in rhizospheric microorganisms under high plant diversity was

increased compared with that under low plant diversity (Lange et al. 2015). Grazing management had

an obvious influence on aboveground biomass, which could explain approximately 50% of the variety

in microbial biomass. These results support the notion that grazing management could change soil microbial activities by regulating ST, SM, and aboveground biomass, which changes the microbial biomass in the soil (Stark et al. 2015; Xu et al. 2017). Overall, the regulation of soil microbial biomass by grazing management is a complex biochemical cycle process, which does not only involve the

control of biological factors, but also includes the regulation of abiotic factors.

*Author contributions.*    Fuiiang Hou designed and directed the study, Zhen Wang carried out the data analysis, and wrote the manuscript. Xiuli Wan, Junbo Chen, Mei Tian, Xiayan Wang,

Xianjiang Chen, Shenghua Chang collected samples, analyzed the data, and contributed to the final writing of the manuscript.

*Data availability.* Data are available from the author, Fujiang Hou (cyhoufj@lzu.edu.cn), upon request.

*Competing interest.*    The authors declare that they have no conflicts of interest.

*Acknowledgements.*    We wish to thank Dr. Cory Matthew and Roxanne Henwood for the help in the revision of the manuscript. We are grateful to Feifei Shi and Lingyu Yang for their help in making Figure 1. This work was supported by the Program for Changjiang Scholars and Innovative Research

Team in University (IRT17R50), the National Key Basic Research Program of China (2014CB138706), the National Natural Science Foundation of China (31172249), the Strategic Priority Research Program of Chinese Academy of Sciences (XDA2010010203), and the 111 project (B12002).



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





**Table 1.** Effects of grazing pattern (GP), grazing intensities (GI), year, sampling time , and their interactions on the rates of Rs, SMBC, and SMBN . F- and P-values were obtained by the General Linear Effects Model (GLE). Significance levels of P < 0.05 are indicated in bold.

| | Rs | | | SMBC | | | SMBN | | |
|---|---|---|---|---|---|---|---|---|---|
| | df | F Value | P | df | F Value | P | df | F Value | P |
| GP | 1 | 116.67 | **<0.001** | 1 | 8.46 | **0.004** | 1 | 3.67 | 0.057 |
| GI | 3 | 2.79 | **0.04** | 3 | 4.06 | **0.008** | 3 | 1.19 | 0.315 |
| Year | 1 | 295.75 | **<0.001** | 2 | 266.32 | **<0.001** | 3 | 5.82 | **0.001** |
| Sampling time | 1 | 375.52 | **<0.001** | 2 | 56.67 | **<0.001** | 2 | 30.03 | **<0.001** |
| GP * Year | 1 | 254.29 | **<0.001** | 2 | 9.98 | **<0.001** | 3 | 0.13 | 0.942 |
| GP* Sampling time | 1 | 45.90 | **<0.001** | 2 | 6.63 | **0.002** | 2 | 0.27 | 0.762 |
| GP * GI | 3 | 11.32 | **<0.001** | 3 | 1.28 | 0.283 | 3 | 1.83 | 0.143 |
| Year * GI | 3 | 18.85 | **<0.001** | 6 | 1.13 | 0.346 | 9 | 1.39 | 0.197 |
| Sampling time * GI | 3 | 9.93 | **<0.001** | 6 | 0.66 | 0.686 | 6 | 1.03 | 0.405 |
| Year * Sampling time | 1 | 133.02 | **<0.001** | 2 | 4.01 | 0.019 | 2 | 6.20 | **0.002** |





**Table 2.** Temperature sensitivity of soil respiration ($Q_{10}$) for the grazing patterns with different grazing intensities (0, 2.7, 5.3, 8.7 sheep $ha^{-1}$).

| GP | GI (sheep $ha^{-1}$) | a | b | $r^2_{adj}$ | p | $Q_{10}$ |
|---|---|---|---|---|---|---|
| WG | 0 | 0.510 | 0.027 | 0.212 | **<0.001** | 1.310 |
| | 2.7 | 0.318 | 0.045 | 0.478 | **<0.001** | 1.568 |
| | 5.3 | 0.313 | 0.043 | 0.458 | **<0.001** | 1.537 |
| | 8.7 | 0.343 | 0.039 | 0.373 | **<0.001** | 1.477 |
| CG | 0 | 0.473 | 0.031 | 0.271 | **<0.001** | 1.363 |
| | 2.7 | 0.689 | 0.019 | 0.155 | **<0.001** | 1.209 |
| | 5.3 | 0.643 | 0.022 | 0.200 | **<0.001** | 1.246 |
| | 8.7 | 0.598 | 0.023 | 0.196 | **<0.001** | 1.259 |

a and b are two coefficients in the regression line $Rs = ae^{bT}$, where Rs is soil respiration and T is soil temperature. $r^2_{adj}$ is the adjustive determinant coefficient, Q10 is the temperature quotient ($=\hat{} e^{10b}$)

5  Significance levels of $P < 0.05$ are indicated in bold.





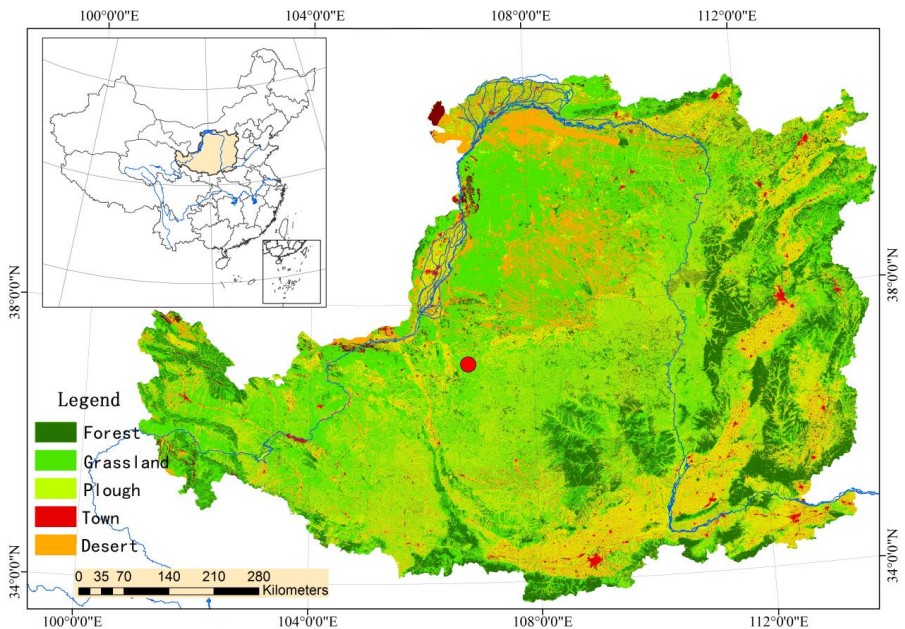

**Figure 1.** Location of the field site on the Loess Plateau. Red circle represents the location of the grazing experiment.





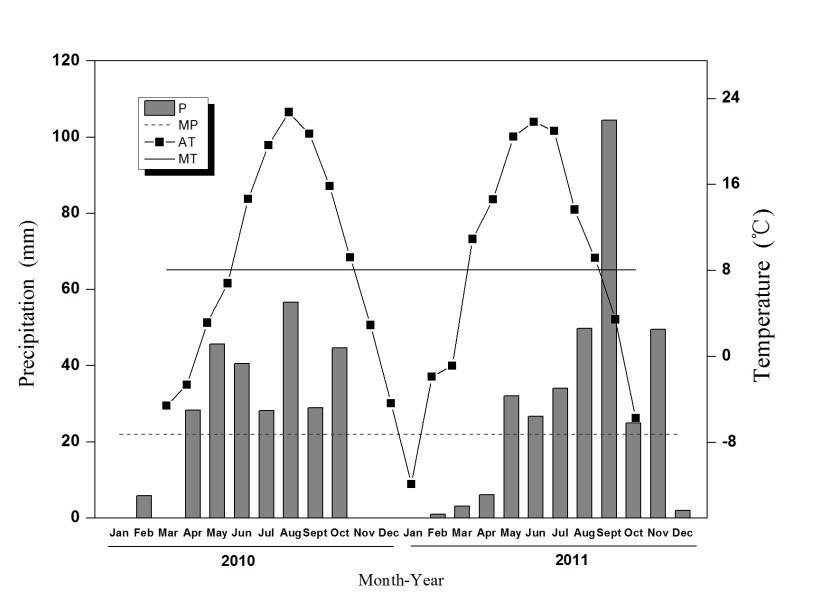

**Figure. 2.** Temporal variation in measured values of air temperature, precipitation, at the study site from January 2010 to December 2011. Straight line and line of dashes represents mean annual precipitation represents and mean annual air temperature from 2001 to 2009. The vegetation of the grassland starts to regreen in late April to early May, and starts to wither in late October.

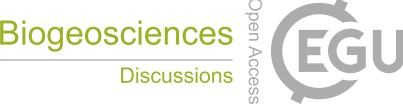

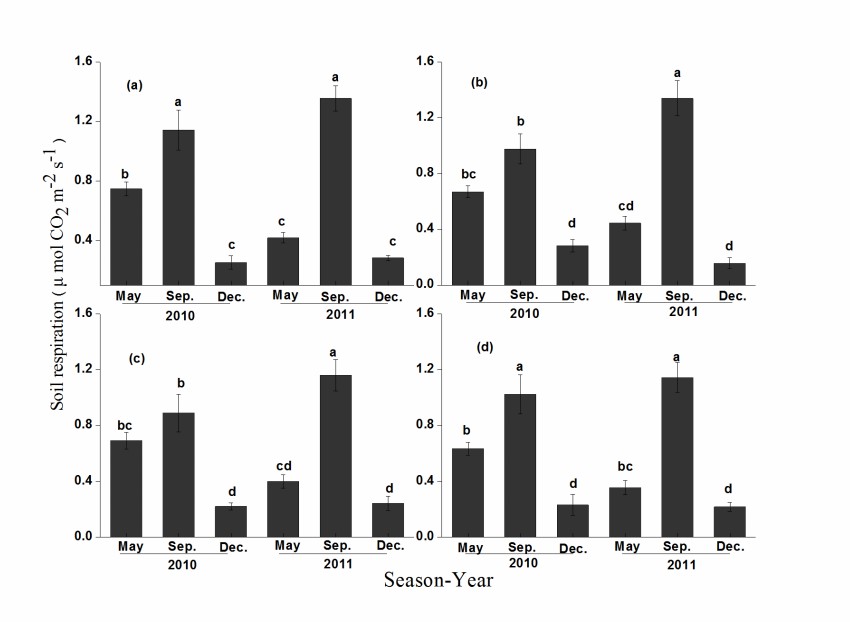

**Figure. 3.** Seasonal dynamics of Rs with different grazing intensities (a) 0 sheep ha$^{-1}$; (b) 2.7 sheep ha$^{-1}$;

(c) 5.3 sheep ha$^{-1}$; (4) 8.7 sheep ha$^{-1}$ in warm season grazing plots. Vertical bars indicate mean bars

standard errors for nine replicates; different lowercase letters indicate a significant difference between

5    month while the same letters indicate no statistical difference ($P < 0.05$).





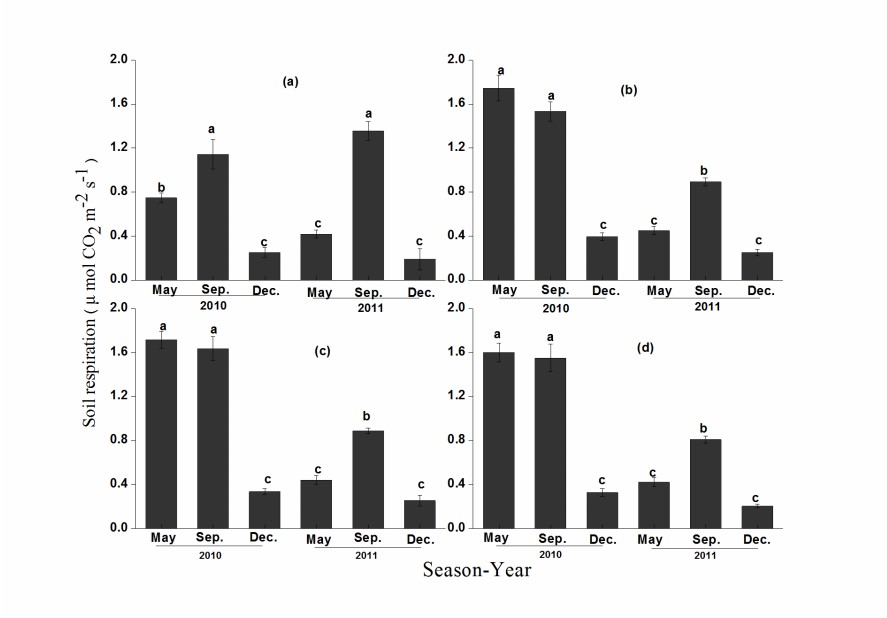

**Figure. 4.** Seasonal dynamics of Rs with different grazing intensities (a) 0 sheep ha-1; (b) 2.7 sheep

ha-1; (c) 5.3 sheep ha-1; (4) 8.7 sheep ha-1 in cold season grazing plots. Vertical bars indicate mean

5    bars standard errors for nine replicates; different lowercase letters indicate a significant difference

between month while the same letters indicate no statistical difference ($P < 0.05$).





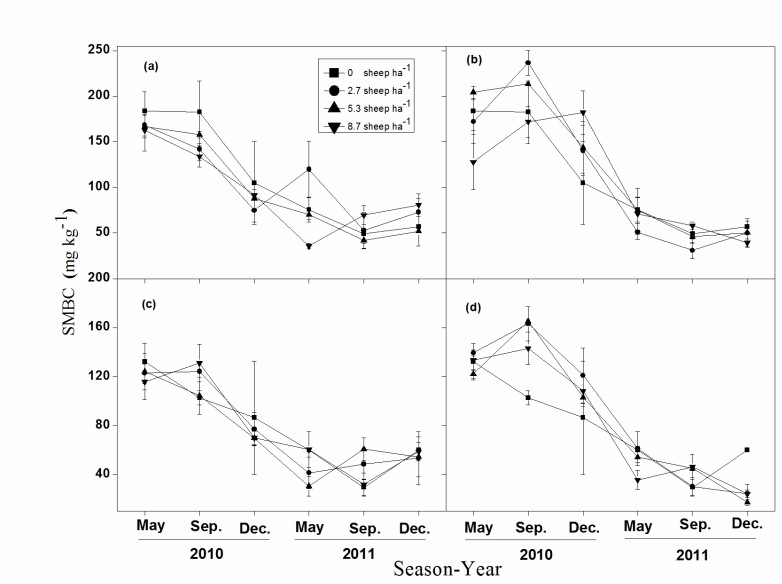

**Figure. 5.** Seasonal dynamics of Soil microbial biomass carbon (SMBC) (a) 0-5 cm soil layers in warm

season grazing plots; (b) 0-5 cm in cold season grazing plots; (c) 5-10 cm in warm season grazing plots;

(d) 5-10 cm in warm season grazing plots .Vertical bars represent the standard error of the measurement

5    mean (n =3) for each observation date.




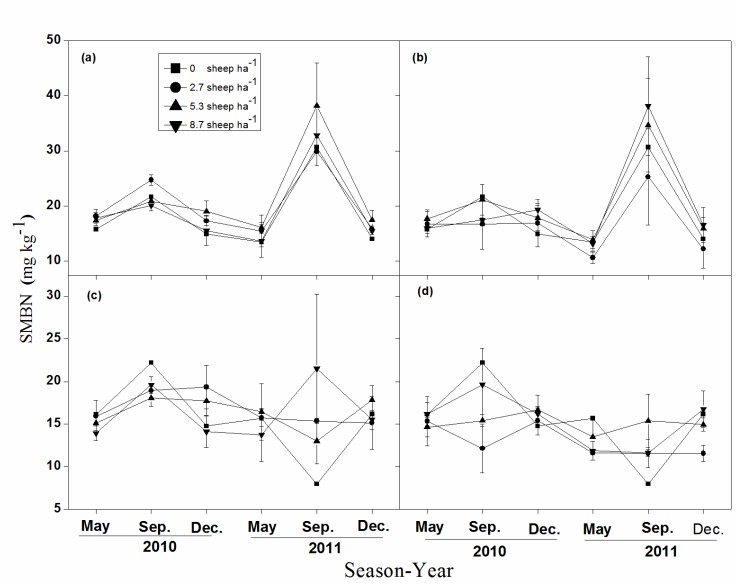

**Figure. 6.** Seasonal dynamics of soil microbial biomass nitrogen (SMBN)(a) 0-5 cm in warm season grazing plots; (b) 0-5 cm in cold season grazing plots; (c) 5-10 cm in warm season grazing plots; (d) 5-10 cm in warm season grazing plots .Vertical bars represent the standard error of the measurement

5    mean (n =3) for each observation date.



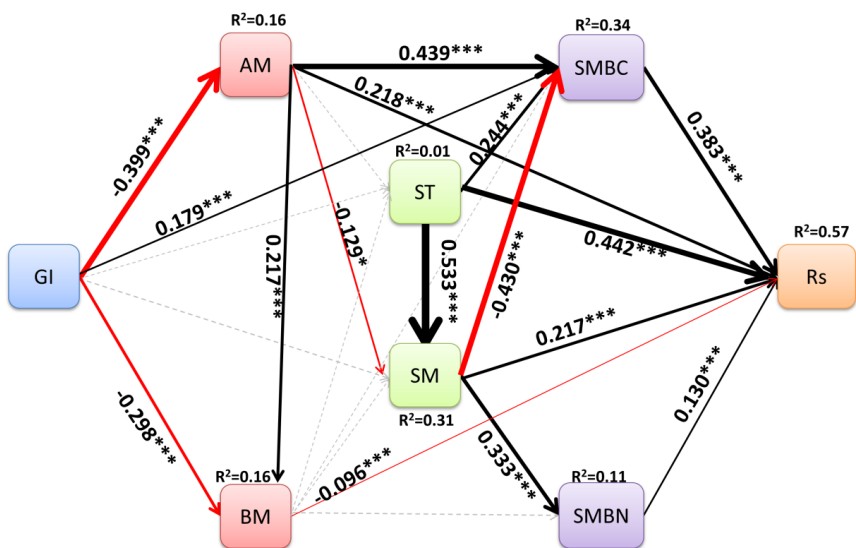

**Figure. 7.** A structural equation model of grazing intensities (GI) effects on soil respiration (Rs). AM, BM, ST, SM, SMBC, SMBN represent aboveground biomass, belowground biomass, soil temperature, soil moisture, soil microbial biomass carbon and soil microbial biomass nitrogen, respectively. The structural equation model considered all plausible pathways through which experimental treatments influence Rs. Red and black arrows represent significant negative and positive pathways, respectively. Bold numbers indicate the standard path coefficients. Arrow width is proportional to the strength of the relationship. $R^2$ represent the proportion of variance explained for each dependent variable in the model. ***$P<0.001$, **$P<0.01$, *$<0.05$; $\chi=10.746$; $P=0.057$; root mean square error of approximation (RMSEA) $=0.059$; $P=0.057$; Akaike information criteria (AIC) $=88.746$.





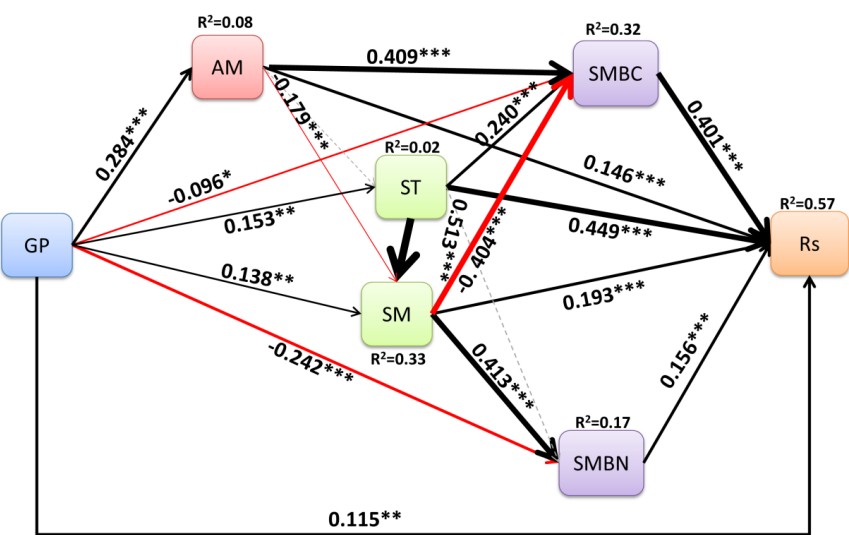

**Figure. 8.**     A structural equation model of grazing patterns (GP) effects on soil respiration (Rs). AM,

ST, SM, SMBC, SMBN represent aboveground biomass, soil temperature, soil moisture, soil microbial

biomass carbon and soil microbial biomass nitrogen, respectively. The structural equation model

5     considered all plausible pathways through which experimental treatments influence Rs. Red and black

arrows represent significant negative and positive pathways, respectively. Bold numbers indicate the

standard path coefficients. Arrow width is proportional to the strength of the relationship. $R^2$ represent

the proportion of variance explained for each dependent variable in the model. ***$P<0.001$, **$P<0.01$,

*$<0.05$; $\chi=2.418$; $P=0.299$; root mean square error of approximation (RMSEA) $=0.025$; $P =0.299$;

10     Akaike information criteria (AIC) $=68.418$.

