# Peer review of "Response of soil respiration and soil microbial biomass carbon and nitrogen to grazing management in the Loess Plateau, China"

_Biogeosciences, 2018_

## Referee Comment (RC1) · Anonymous Referee #1 · 17 Jan 2019

Dear editor, I reviewed this manuscript entitled "Response of soil respiration and soil microbial biomass carbon and nitrogen to grazing management in the Loess Plateau, China "which analyzed soil respiration, SMBC, and SMBN under different grazing intensities and seasonal grazing patterns (in summer or winter) , also abiotic and biotic factors were measured. The experimental design is reasonable, the indicators and data collection are sufficient, the analysis and demonstration are rigorous, the quantity and quality of charts are reasonable, and this research has important theoretical and practical significance. Basing on the above comments, the manuscript is fitted to the standard of SCI journal, but there are some problems has existed , so I suggest the author revise the manuscript before publishing it. Now I give some problems detailedly

as follows. Page1 L18-19 What are the specific rotational grazing methods in the 9-year grazing areas mentioned here? Are there any combinations in the design of the experiment? Page4 L26-27 Soil respiration is significantly affected by soil moisture and temperature, and whether the particularity of precipitation in September 2011 has an impact on soil respiration data measured in September? Page5 L2 Is there any reference or self-setting in the classification of grazing intensity? Please indicate the basis. Page5 L17-18 Diurnal and seasonal variations of soil respiration were different. Did the authors consider them during the measurement period? Page7 L15-17 The author can analyze the daily changes of soil respiration of winter grazing and summer grazing under the same grazing gradient to determine which grazing intensity and grazing mode has a great influence. Page8 L5 It is suggested that divide this part into two. The first part is to analyze the change of SMBC and SMBN under different grazing intensity and grazing mode, and the second part is to analyze the interaction between abiotic and biotic factors and structural equation model. Page9 L27-28 The author has discussed enough. The small suggestion is that soil respiration is closely related to the change of A, and the change of the relationship between the two under different grazing intensity and grazing mode can be considered. Figures Fig. 2 The significance test results are not prominent, suggesting recommend the results which is marked significantly and omit unsignificantly. Fig. 5-6 "(d) 5-10 cm in warm season grazing plots" should change to "(d) 5-10 cm in cold season grazing plots" in both Fig. 5and Fig. 6.

---

## Referee Comment (RC2) · Anonymous Referee #2 · 10 Feb 2019

The authors investigated how grazing intensities and grazing patterns affect soil respiration as well as the potential underlying mechanisms. Their results are interesting and can be potentially published somewhere. But for the current MS, it was quite confusion and unclear (please see some of my specific comments below). Moreover, there were many gramma errors across the whole MS, I did not list all of these errors, it was quite time-consuming. I deeply understand the writing difficulties for the non-native English researchers, but this MS was quite immature for submission. I suggest that authors should well prepare their manuscript for the next submission.

Abstract The abstract should be rewritten, particularly for the description of your re-

sults. You should focus on what you are really want to let others know from this study. The conclusion from the results was quite vague, nothing valuable from your conclusion sentence. You have a lot of information about your experiments design, is it too specific in the abstract section? Can you describe your experiment design in a much terser way? P1 L16-18. Really? I think there are already many studies investigating these variables, even several meta-analysis. You should reorganize the sentence. P1 L 25. I think the word "affect" in your results section is very vague. Readers will not know whether grazing increase or decrease Rs in this way. P2 L1-2. Quite confusion. Are these results from SEM analysis? Or you just add this sentence. P2 L4. Nothing new from the last sentence of your abstract. I think as a researcher in grazing ecology, one can easily hypothesized that grazing can affect C sequestration through both biotic and abiotic factors. Do you think your conclusion is very new and deserve to be published? One suggestion was that you should be very specific, then you will have your own conclusion. Introduction At the beginning, I should highlight that there are many gramma errors in your whole manuscript; I will not list your errors one by one. I think this is your work, which should be finished before your submission. You have too many abbreviations. It is quite difficult for me to remember so many abbreviations, I need to refresh these abbreviations frequently. Moreover, many abbreviations only appeared once or twice. P2 L15. A repetition of the first sentences in your abstract. P2 L22. You have a good literature review. Then what are your research questions? The second paragraph of your introduction was very long, however, you changed your logical and focus for several times. It is very hard for readers to understand what you want to say. You should sharp your research questions and hypotheses. In your current version, you always present your hypotheses, questions and results in quite ambiguous ways. It is quite difficult for readers to find something interesting from your MS, if you determine present your results in this way. Materials and methods P5 L10. Why do you only measure soil respiration in 2010 and 2011, considering you have conducted this experiment for nine years? P5 L10. Do you continue with the measurements from 2011-2018, since these experiments were conducted eight years ago? The description about your Rs measurements is unclear, even though you had some citations here. Why do you only measure Rs during the middle of May, September, and December? Do you mean ST and SM measured for all treatments or only for the control? It was quite confusion. Do you mean ST and SM only measure for the dates when RS was measured? How can you come true the random but adjacent the pots for Rs measurements? As soon as possible? How fast is it? Within several minutes, several hours or even several days? Many variables were measured repeatedly across the seasons or years, so a repeated ANOVA analysis should be used for your statistical analysis. You should have more information about the description of SEM analysis. Results Your whole results sections are quite confusion. I think there are many related studies published, you can read how they write their results section. After the subtitle "SMBC and SMBN", you have a lot of description on ST, SM, AGB, BGB, or even the results from your data analysis. Do you need more subtitles? Discussion Sorry, I do not read your discussion. There are many gramma errors, confusion sentences or even very strange descriptions impeding my review. I will stop here. I think the authors should well prepare their manuscript for the next submission. Figures and tables Table 1. Repeated ANOVA analysis should be conducted. Table 2. What do "WG" and "CG" stand for? Figure 1. Is this figure related to your study? You have legend for forest, grassland. . . If you want show a figure like this, I would suggest you show your experiments design since it was now very confusion. Figure 3. Why did you only measure soil respiration from three months? Figures should be presented in a easier way depending on what you want to compare. Figure 5. Why do your determine to use line chart here? There are many overlaps. You symbols are not very similar. It was very hard for me to understand your figure. Abbreviations were rarely used in the titles or in the first word of a sentence. How can you construct your SEM in this way? Was it based on your model comparison or randomly?

---

## Author Comment (AC1) · 18 Mar 2019

Response to Anonymous Referee #2

General comments:

The authors investigated how grazing intensities and grazing patterns affect soil respiration as well as the potential underlying mechanisms. Their results are interesting and can be potentially published somewhere. But for the current MS, it was quite confusion and unclear (please see some of my specific comments below). Moreover, there were many gramma errors across the whole MS, I did not list all of these errors, it was quite

time-consuming. I deeply understand the writing difficulties for the non-native English researchers, but this MS was quite immature for submission. I suggest that authors should well prepare their manuscript for the next submission.

Author's response: Dear Referee, first of all, we would like to thank you for the time you devoted to reviewing this manuscript. We explain how we have revised the manuscript by following the referee's comments one by one. We have made major revisions on the Abstract, Introduction, Discussion and Conclusions. The revised manuscript has been polished by native English speaker, Mr. Roger Lucien Daives.

Abstract The abstract should be rewritten, particularly for the description of your results. You should focus on what you are really want to let others know from this study.

Author's response: We truly appreciate your constructive suggestions. Changes were made following your advices. We had been rewritten the abstract in the new revisions. Please see P 1 L 15 .

You have a lot of information about your experiments design, is it too specific in the abstract section? Can you describe your experiment design in a much terser way?

Author's response: Thanks. We have rewritten the experiment design in the abstract section in the revised version. Please see P 1 L21-23.

P1 L16-18. Really? I think there are already many studies investigating these variables, even several meta-analysis. You should reorganize the sentence.

Author's response: Thanks. We have re-worded this sentence as "Grazing management affects grassland carbon dynamics and soil microbial biomass, yet the effects of grazing management, such as grazing intensity and grazing regimes (GP), on soil respiration and soil microbial biomass carbon (SMBC) and nitrogen (SMBN) are not fully understood", see P1 L 15-18.

L 25. I think the word "affect" in your results section is very vague. Readers will not know whether grazing increase or decrease Rs in this way.

Author's response: Thanks for your constructive suggestions, we have revised this sentence. Please see P 1 L25

P2 L4. Nothing new from the last sentence of your abstract. I think as a researcher in grazing ecology, one can easily hypothesized that grazing can affect C sequestration through both biotic and abiotic factors. Do you think your conclusion is very new and deserve to be published? One suggestion was that you should be very specific, then you will have your own conclusion.

Author's response: We appreciate the referee's suggestion. To make these sentences clear, we improved them as "Field experimental results indicate that the effects of grazing management on Rs and other soil microbial carbon and nitrogen processes in the grazing system depend on more grazing regime (GP) than grazing intensity; and suggest that GP should be considered in future manipulation experiments and included in carbon models to accurately simulate soil carbon dynamics under scenarios of climate change in grassland ecosystems", see P 2 L 6.

Introduction

At the beginning, I should highlight that there are many gramma errors in your whole manuscript; I will not list your errors one by one. I think this is your work, which should be finished before your submission.

Author's response: Sorry, our previous MS caused the misunderstanding by the referee. Roger L. Davies (NZCFS) has assisted in editing this research paper.

You have too many abbreviations. It is quite difficult for me to remember so many abbreviations, I need to refresh these abbreviations frequently. Moreover, many abbreviations only appeared once or twice.

Author's response: We apologize for the confusion resulting from the unclear statements in the manuscript. We have deleted some unnecessary abbreviations in the revised manuscript.

P2 L15. A repetition of the first sentences in your abstract

Author's response: Thanks, We have changed this sentence as "Grassland covers about 41% of earth's terrestrial surface. . ." in the revised revision.

P2 L22. You have a good literature review. Then what are your research questions? The second paragraph of your introduction was very long, however, you changed your logical and focus for several times. It is very hard for readers to understand what you want to say. You should sharp your research questions and hypotheses. In your current version, you always present your hypotheses, questions and results in quite ambiguous ways. It is quite difficult for readers to find something interesting from your MS, if you determine present your results in this way.

Author's response: Thanks. We have rewritten the second paragraph in the revised manuscript. Please see P 3 L 2.

Materials and methods P5 L10. Why do you only measure soil respiration in 2010 and 2011, considering you have conducted this experiment for nine years? Do you continue with the measurements from 2011-2018, since these experiments were conducted eight years ago? The description about your Rs measurements is unclear, even though you had some citations here.

Author's response: Thanks for your comments. The rotational grazing experiment began in 2001, and the Rs and soil microbial C and N measurements were carried out from 2010 to 2011, the previous 9 years of field trials being used for other experimental purposes. We continue with the measurements from 2012-2018. Since 2012, we have added another experimental design on the original basis. For this reason, the data beyond 2012 is not shown in this manuscript. We have rewritten the description about Rs measurements in the revised manuscript. Please see P 5 L 21.

Why do you only measure Rs during the middle of May, September, and December?

Author's response: Thanks. The Loess Plateau belongs to temperate continental monsoon climate. The Loess Plateau belongs to temperate continental monsoon climate. The early stage of herbage growth begins in mid May; aboveground grassland biomass peaked in mid September; grassland dormancy occurs by mid December. For these reasons, Rs was measured in mid May, September, and December. Please see P 5 L24.

Do you mean ST and SM measured for all treatments or only for the control? It was quite confusion.

Author's response: Thank you for pointing out that this was confusing. Soil temperature and soil moisture were measured for all treatments. We have changed this sentence as "Soil temperature at 10 cm depth was measured in real time using a thermocouple probe attached to every soil efflux collar during each Rs measurement. Soil moisture samples for gravimetric analysis were taken from the top 10.0 cm, close to every soil efflux collar, once a day in mid morning morn of Rs sampling days, and oven dried at 105°C for 48 h " . Please see P6 L5.

Do you mean ST and SM only measure for the dates when RS was measured? How can you come true the random but adjacent the pots for Rs measurements? As soon as possible? How fast is it? Within several minutes, several hours or even several days?

Author's response: We are thankful for the reviewer's suggestion. Soil temperature at 10 cm depth was measured immediately using a thermocouple probe which attached to the gas analyzer adjacent to each PVC collar during the time of Rs measurement for all treatment. Soil moisture samples for gravimetric analysis were taken from the top 10.0 cm, close to every soil efflux collar, once a day in mid mornin morning of Rs sampling days. Rs measurement by gas analyzer took approximately 9 min to complete per plot. Please see P 5 L 20.

Many variables were measured repeatedly across the seasons or years, so a repeated ANOVA analysis should be used for your statistical analysis

Author's response: Thanks. A repeated ANOVA analysis have been conducted in the revised manuscript. Please see P 7 L4.

You should have more information about the description of SEM analysis.

Author's response: Thanks. As you suggested, we have added detailed information for SEM analysis. See P7 L20, and Supplementary Figure 1. Results

Your whole results sections are quite confusion. I think there are many related studies published, you can read how they write their results section.

Author's response: We appreciate the reviewer's comments that could help us improve our manuscript. We have rewritten results sections in the new version of manuscript.

After the subtitle "SMBC and SMBN", you have a lot of description on ST, SM, AGB, BGB, or even the results from your data analysis. Do you need more subtitles?

Author's response: Thanks for your constructive suggestion. We added new subtitles as "3.3 Effects of grazing management on soil temperature, soil moisture, aboveground biomass, belowground biomass"; "3.4 Effect of grzaing management on temperature sensitivity of soil respiration"; "3.5 Structural equation models" in the revised manuscript. Please see P 9 L17.

Discussion Sorry, I do not read your discussion. There are many gramma errors, confusion sentences or even very strange descriptions impeding my review. I will stop here. I think the authors should well prepare their manuscript for the next submission.

Author's response: In the new version, we rewrote the discussion section and deleted sections to avoid the misunderstanding. Roger L. Davies (NZCFS) has assisted in editing this research paper.

Figures and tables Table 1. Repeated ANOVA analysis should be conducted.

Author's response: A repeated ANOVA analysis have been conducted in the revised manuscript. Please see P 24 L1.

Table 2. What do "WG" and "CG" stand for?

Author's response: "WG" stand for warm season grazing ; "CG" stand for cold season grazing. Please see lines P21 L16 in the revised manuscript.

Figure 1. Is this figure related to your study? You have legend for forest, grassland. . .If you want show a figure like this, I would suggest you show your experiments design since it was now very confusion.

Author's response: Thanks. We have deleted this figure and added a new figure about our experiments design. Please see P 27 L 3.

Figure 3. Why did you only measure soil respiration from three months? Figures should be presented in a easier way depending on what you want to compare.

Author's response: The Loess Plateau belongs to temperate continental monsoon climate. The early stage of herbage growth begins in mid May; aboveground grassland biomass peaked in mid September; grassland dormancy occurs by mid December. For these reasons, Rs was measured in mid May, September, and December. To make better understand the figure, we merged Figures 3 and 4 into one figure, and then we made bar graph into thick color lines. Please see P 5 L24

Figure 5. Why do your determine to use line chart here? There are many overlaps. You symbols are not very similar. It was very hard for me to understand your figure.

Author's response: We apologize for the confusion from the chart. In order to show seasonal variations of soil microbial mass carbon and nitrogen at 5 cm and 10 cm soil depth in the warm grazing and cold grazing grassland under different grazing intensity, for this reason, we used line chart here. To make better understand the chart, we made black lines into thick color lines and added more information in chart. Please see P 30 L 1

Abbreviations were rarely used in the titles or in the first word of a sentence. How can you construct your SEM in this way? Was it based on your model comparison or

randomly?

Author's response: We appreciate the referee's suggestion. We have made these points clear in the revised manuscript. SEM analysis was conducted according to a Priori conceptual model, to include all possible pathways (Supplementary Figure 1), including (1) both direct and indirect pathways of GP and GI on aboveground biomass, belowground biomass, soil temperature, soil moisture; (2) both direct and indirect pathways of GP and GI on soil microbial carbon and nitrogen; (3) both direct and indirect pathways of GP or GI on Rs via biotic or abiotic factors. To differentiate the effects of grazing management on Rs, grazing management was divided into two sections. The first SEM focuses on the direct or indirect effect of GI on Rs; the second SEM focuses on both direct and indirect effect of GP on Rs. Plesae See P 7 L 20.

Please also note the supplement to this comment:
https://www.biogeosciences-discuss.net/bg-2018-531/bg-2018-531-AC1-supplement.pdf

**Supplement:**

**Grazing rate Calculation**

The classification of grazing intensity (GI) was based on local herbage productivity and daily hay intake per sheep unit method grazing intensity calculation use the following formula published by agriculture industry standard NY/T 635–2002 (Ministry of Agriculture of the People's Republic of China):

$$\text{GI(sheep } ha^{-1}) = \frac{A \times Y \times R}{D \times B}$$

Where A is available rangeland area (ha), Y is edible forage yield (kg ha$^{-1}$), R is proper utilization rate of rangeland (%), D is days of grazing (d), B is daily hay intake per sheep unit (kg d$^{-1}$).

**Supplementary Table 1.** Daliy variation in measured values of soil respiration (Rs) at the warm grazing site from May 2010 to December 2011 with different grazing intensities (GI).

| Year | Month | GI (sheep ha$^{-1}$) | Rs ($\mu$mol CO$_2$ m$^{-2}$s$^{-1}$) | | | | | Daily Total C efflux (g CO$_2$ m$^{-2}$d$^{-1}$) |
|------|-------|------|---------|------|---------|------|-----------|-----------|
| | | | Maximum | time | Minimum | time | Mean ±SE | |
| | | 0 | 0.90 | 10:00 | 0.48 | 6:00 | 0.75 ±0.06 | 0.77 |
| | May | 2.7 | 0.86 | 10:00 | 0.49 | 6:00 | 0.67 ±0.04 | 0.69 |
| | | 5.3 | 1.02 | 10:00 | 0.41 | 6:00 | 0.69 ±0.06 | 0.72 |
| | | 8.7 | 0.82 | 10:00 | 0.39 | 6:00 | 0.63 ±0.03 | 0.65 |
| | | 0 | 1.69 | 14:00 | 0.60 | 22:00 | 1.16 ±0.05 | 1.20 |
| 2010 | Sep | 2.7 | 1.38 | 14:00 | 0.54 | 22:00 | 0.99 ±0.08 | 1.03 |
| | | 5.3 | 1.41 | 14:00 | 0.49 | 22:00 | 0.91 ±0.09 | 0.94 |
| | | 8.7 | 1.62 | 12:00 | 0.45 | 22:00 | 1.04 ±0.09 | 1.08 |
| | | 0 | 0.32 | 14:00 | 0.16 | 6:00 | 0.25 ±0.04 | 0.26 |
| | Dec | 2.7 | 0.37 | 14:00 | 0.23 | 6:00 | 0.28 ±0.04 | 0.29 |
| | | 5.3 | 0.25 | 14:00 | 0.17 | 6:00 | 0.22 ±0.03 | 0.23 |
| | | 8.7 | 0.360 | 14:00 | 0.099 | 6:00 | 0.23 ±0.08 | 0.24 |
| | | 0 | 0.55 | 10:00 | 0.31 | 22:00 | 0.42 ±0.05 | 0.44 |
| | May | 2.7 | 0.68 | 10:00 | 0.30 | 6:00 | 0.44 ±0.04 | 0.46 |
| | | 5.3 | 0.58 | 10:00 | 0.21 | 6:00 | 0.39 ±0.04 | 0.40 |
| | | 8.7 | 0.58 | 10:00 | 0.15 | 6:00 | 0.35 ±0.03 | 0.36 |
| | | 0 | 1.69 | 16:00 | 1.02 | 22:00 | 1.35 ±0.08 | 1.40 |
| 2011 | Sep | 2.7 | 1.84 | 16:00 | 0.81 | 6:00 | 1.34 ±0.09 | 1.39 |
| | | 5.3 | 1.58 | 14;00 | 0.69 | 6:00 | 1.16 ±0.09 | 1.20 |
| | | 8.7 | 1.62 | 16:00 | 0.78 | 6:00 | 1.14 ±0.06 | 1.18 |
| | | 0 | 0.30 | 14:00 | 0.16 | 6:00 | 0.24 ±0.04 | 0.25 |
| | Dec | 2.7 | 0.22 | 14:00 | 0.084 | 6:00 | 0.16 ±0.04 | 0.16 |
| | | 5.3 | 0.34 | 14:00 | 0.185 | 6:00 | 0.24 ±0.05 | 0.25 |
| | | 8.7 | 0.27 | 14:00 | 0.16 | 6:00 | 0.22 ±0.03 | 0.22 |

**Supplementary Table 2.** Daliy variation in measured values of soil respiration (Rs) at the cold grazing site from May 2010 to December 2011 with different grazing intensities (GI).

| Year | Season | GI (sheep ha$^{-1}$) | Rs ($\mu$mol $CO_2$ m$^{-2}$ s$^{-1}$) | | | | | Daily Total C efflux (g $CO_2$ m$^{-2}$ d$^{-1}$) |
|------|--------|------|---------|------|---------|------|-----------|--------|
| | | | Maximum | time | Minimum | time | Average $\pm$ SE | |
| | | 0 | 0.90 | 10:00 | 0.48 | 6:00 | 0.75 ±0.06 | 0.77 |
| | May | 2.7 | 2.17 | 16:00 | 1.35 | 22:00 | 1.74 ±0.08 | 1.81 |
| | | 5.3 | 1.90 | 16:00 | 1.27 | 6:00 | 1.71 ±0.05 | 1.78 |
| | | 8.7 | 1.87 | 16:00 | 1.13 | 6:00 | 1.60 ±0.05 | 1.66 |
| | | 0 | 1.69 | 14:00 | 0.60 | 22:00 | 1.16 ±0.11 | 1.20 |
| 2010 | Sep | 2.7 | 1.86 | 14:00 | 1.23 | 6:00 | 1.53 ±0.05 | 1.59 |
| | | 5.3 | 2.11 | 16:00 | 1.18 | 22:00 | 1.63 ±0.06 | 1.69 |
| | | 8.7 | 2.02 | 14:00 | 1.07 | 6:00 | 1.55 ±0.09 | 1.61 |
| | | 0 | 0.32 | 14:00 | 0.16 | 6:00 | 0.25 ±0.04 | 0.26 |
| | Dec | 2.7 | 0.46 | 14:00 | 0.34 | 6:00 | 0.39 ±0.04 | 0.41 |
| | | 5.3 | 0.38 | 14:00 | 0.30 | 6:00 | 0.34 ±0.02 | 0.35 |
| | | 8.7 | 0.37 | 14:00 | 0.26 | 6:00 | 0.33 ±0.03 | 0.34 |
| | | 0 | 0.55 | 10:00 | 0.31 | 22:00 | 0.42 ±0.05 | 0.43 |
| | May | 2.7 | 0.69 | 16:00 | 0.31 | 6:00 | 0.45 ±0,03 | 0.47 |
| | | 5.3 | 0.61 | 10:00 | 0.23 | 22:00 | 0.44 ±0.03 | 0.46 |
| | | 8.7 | 0.61 | 10:00 | 0.25 | 6:00 | 0.42 ±0.03 | 0.44 |
| | | 0 | 1.69 | 16:00 | 1.02 | 22:00 | 1.35 ±0.08 | 1.40 |
| 2011 | Sep | 2.7 | 1.02 | 16:00 | 0.72 | 6:00 | 0.89 ±0.03 | 0.93 |
| | | 5.3 | 1.01 | 10:00 | 0.77 | 6:00 | 0.89 ±0.03 | 0.92 |
| | | 8.7 | 0.95 | 10:00 | 0.68 | 22:00 | 0.81 ±0.03 | 0.84 |
| | | 0 | 0.30 | 14:00 | 0.16 | 6:00 | 0.24 ±0.04 | 0.25 |
| | Dec | 2.7 | 0.28 | 14:00 | 0.19 | 6:00 | 0.25 ±0.03 | 0.26 |
| | | 5.3 | 0.34 | 14:00 | 0.17 | 6:00 | 0.25 ±0.05 | 0.26 |
| | | 8.7 | 0.23 | 14:00 | 0.18 | 6:00 | 0.21 ±0.02 | 0.21 |

10

 **Supplementary Figure 1**  An a priori conceptual model of how grazing intensity (GI), grzing regime (GP) altered soil respiration (Rs), soil microbial carbon and nitrogen. The model contains all possible pathways that cause changes in the abiotic and biotic variables and soil respiration. Box represents variables Arrow direction indicates the hypothesized direction of causation.

**Supplementary Figure 2**    Seasonal dynamics of soil temperature (a) in warm season grazing plots; (b) in cold season grazing plots; Vertical bars represent the standard error of the measurement mean (n=9) for each observation date.

10

15

**Supplementary Figure 3**    Seasonal dynamics of soil moisture (a) in warm season grazing plots; (b) in cold season grazing plots; Vertical bars represent the standard error of the measurement mean (n =3) for each observation date.

10

15

20

25

**Supplementary Figure 4.** Aboveground biomass within (a) warm season grazing plots; (b) cold season grazing plots; belowground biomass within (c) warm season grazing plots; (d) cold season grazing plots from 2010 to 2011. Horizontal lines in boxes show medians and dashed whiskers show data extremes. Open circles and solid whiskers show means ±propagated standard errors.

---

## Author Comment (AC2) · 18 Mar 2019

Response to Anonymous Referee #1

General comments:

Dear editor, I reviewed this manuscript entitled "Response of soil respiration and soil microbial biomass carbon and nitrogen to grazing management in the Loess Plateau, China "which analyzed soil respiration, SMBC, and SMBN under different grazing intensities and seasonal grazing patterns (in summer or winter) , also abiotic and biotic factors were measured. The experimental design is reasonable, the indicators and

data collection are sufficient, the analysis and demonstration are rigorous, the quantity and quality of charts are reasonable, and this research has important theoretical and practical significance. Basing on the above comments, the manuscript is fitted to the standard of SCI journal, but there are some problems has existed , so I suggest the author revise the manuscript before publishing it.

Response from authors: We are highly grateful for the reviewer's positive comments on our work. We carefully considered your comments and will take them into account for further revisions.

Page1 L18-19 What are the specific rotational grazing methods in the 9-year grazing areas mentioned here? Are there any combinations in the design of the experiment?

Response from authors: Thanks for these constructive comments. The specific rotational grazing methods are: The experiment of the rotational grazing system started from 2001. We choose two areas which had similar topographic conditions, vegetation composition and cover for warm season (summer) grazing and cold season (winter) grazing, respectively. Each of areas was divided into twelve 0.5 ha enclosed plots comprising three replicates for four gradient of grazing, 0, 4, 8, and 13 sheep, representing stocking rates of 0, 2.7, 5.3, and 8.7 sheep/ha, respectively. Grazing rates in this context being defined as the number of animals allocated to a treatment divided by the combined area of 1.5 ha for the three replicates of each treatment in either grazing season. When sheep were allocated to a treatment, a rotational stocking system was used the same sheep grazed successive replicates rotational. Warm season grazing plots were rotationally grazed from June to September. One cycle of rotational grazing is 30 days with 10 days for grazing and 20 days for rest, three rotations for total 90 days. Cold season grazing started from mid-November to late December. One cycle of rotational grazing is 24 days with 8 days for grazing and 16 days for rest, two rotations for total 48 days. Soil respiration measurements were conducted both warm season (summer) grazing plots and cold season (winter) grazing plots after nine years rotational grazing. The rotational grazing experiment began in 2001, and the Rs and soil

microbial C and N measurements were carried out from 2010 to 2011, the previous 9 years of field trials being used for other experimental purposes. We have revised the abstract in the revised manuscript, see P1 L20.

Page4 L26-27 Soil respiration is significantly affected by soil moisture and temperature, and whether the particularity of precipitation in September 2011 has an impact on soil respiration data measured in September?

Response from authors: We completely agree with the reviewer. We have analysed the relationship between monthly precipitation and soil respiration, soil temperature, soil moisture, soil microbial biomass carbon and soil microbial biomass nitrogen. There is a strong relationship precipitation between and those variables. We added the results in the revised version, please see P 7 L 6; P24 L2.

Page5 L2 Is there any reference or self-setting in the classification of grazing intensity? Please indicate the basis.

Response from authors: We greatly appreciate your thoughtful comments. The classification of grazing intensity (GI) was based on local habitat productivity and daily hay intake per sheep unit method grazing intensity calculation use the following formula published by agriculture industry standard NY/T 635–2002 (Ministry of Agriculture of the People's Republic of China): GI(sheep ha $-1=(A \times Y \times R)/(D \times B)$ Where A is available rangeland area (ha), Y is edible forage yield (kg ha-1), R is proper utilization rate of rangeland (%), D is days of grazing (d), B is daily hay intake per sheep unit (kg d-1). We added the detail of grazing intensity in the manuscript, please see P11 L9

Page5 L17-18 Diurnal and seasonal variations of soil respiration were different. Did the authors consider them during the measurement period?

Response from authors: We appreciate the reviewer's comments. We conducted 22 hour (between 6:00am and 10:00pm, at 2 hour intervals) measurements each day to examine the diurnal soil respiration. The early stage of herbage growth begins

in mid May; aboveground grassland biomass peaked in mid September; grassland dormancy occurs by mid December. For these reasons, Rs was measured in mid May, September, and December. Please see P 5 L24.

Page7 L15-17 The author can analyze the daily changes of soil respiration of winter grazing and summer grazing under the same grazing gradient to determine which grazing intensity and grazing mode has a great influence.

Response from authors: We appreciate the reviewer's comments that could help us improve our manuscript. As reviewer suggested, we analyze the daily changes of soil respiration of winter grazing and summer grazing under the same grazing gradient. We found grazing intensity with 0 sheep ha-1 has a great influence on soil respiration in warm grazing plots; grazing intensity with 2.7 sheep ha-1 and 5.3 sheep ha-1 has a strong impact on soil respiration in cold grazing plots. We added the analysis in the revised manuscript in P 6 L 30, P 23 L2.

Page8 L5 It is suggested that divide this part into two. The first part is to analyze the change of SMBC and SMBN under different grazing intensity and grazing mode, and the second part is to analyze the interaction between abiotic and biotic factors and structural equation model.

Response 6: Thanks for these constructive suggestions. We added new subtitles which were "3.3 Effects of grazing management on soil temperature, soil moisture, aboveground biomass, belowground biomass" "3.4 Effect of grazing management on temperature sensitivity of soil respiration" "3.5 Structural equation models" in the revised manuscript. Please see P 9 L17

Page9 L27-28 The author has discussed enough. The small suggestion is that soil respiration is closely related to the change of A, and the change of the relationship between the two under different grazing intensity and grazing mode can be considered.

Response from authors: We are grateful for the reviewer's suggestion. We have discussed the annual soil respiration between the two under different grazing intensity and grazing mode. Please see P 11 L13; P 11 L20.

Figures Fig. 2 The significance test results are not prominent, suggesting recommend the results which is marked significantly and omit unsignificantly.

Response from authors: We greatly appreciate your constructive suggestions. We added the significance test results in figure 2. Please see P 28 L 6.

Fig. 5-6 "(d) 5-10 cm in warm season grazing plots" should change to "(d) 5-10 cm in cold season grazing plots" in both Fig. 5 and Fig. 6. Response from authors: Changes were made following your advices in the revised version. Please see P 29 L 3.

―――――――――――――――――――――

---

## Author Comment (AC3) · 9 Apr 2019

Response to Anonymous Referee #1

General comments:

Dear editor, I reviewed this manuscript entitled "Response of soil respiration and soil microbial biomass carbon and nitrogen to grazing management in the Loess Plateau, China "which analyzed soil respiration, SMBC, and SMBN under different grazing intensities and seasonal grazing patterns (in summer or winter), also abiotic and biotic factors were measured. The experimental design is reasonable, the indicators and

data collection are sufficient, the analysis and demonstration are rigorous, the quantity and quality of charts are reasonable, and this research has important theoretical and practical significance. Basing on the above comments, the manuscript is fitted to the standard of SCI journal, but there are some problems has existed , so I suggest the author revise the manuscript before publishing it.

Response from authors: We are highly grateful for the reviewer's positive comments on our work. We carefully considered your comments and will take them into account for further revisions.

Page1 L18-19 What are the specific rotational grazing methods in the 9-year grazing areas mentioned here? Are there any combinations in the design of the experiment?

Response from authors: Thanks for these constructive comments. The specific methods used for rotational grazing were: In each grazing season, the wethers were allocated to three replicates for each of the four stocking rates, and rotationally grazed between each replicate plot allocated to that stocking rate. The WG plots were rotationally grazed from June to September (90 days), with a rotation cycle length of 30 days (10 days grazing and 20 days rest) and three rotations. The CG plots were grazed from mid-November to late December (48 days), with a rotation cycle length of 24 days (eight days grazing and 16 days rest) and two rotations. The rotational grazing system field experiment began in 2001. The Rs measurements were conducted in both WG and CG plots in 2010, after the trial site had been rotationally grazed for the previous 9 years. We have revised the abstract in the revised manuscript and added the specific rotational grazing methods in revised manuscript. Please see P1 L137-L143.

Page4 L26-27 Soil respiration is significantly affected by soil moisture and temperature, and whether the particularity of precipitation in September 2011 has an impact on soil respiration data measured in September?

Response from authors: We completely agree with the reviewer. We have analysed the relationship between monthly precipitation and soil respiration, soil temperature, soil moisture, soil microbial biomass carbon and soil microbial biomass nitrogen. There is a strong relationship precipitation between and those variables. We added the results in the revised manuscript, please see L193-L195 and L667-L669.

Page5 L2 Is there any reference or self-setting in the classification of grazing intensity? Please indicate the basis.

Response from authors: We greatly appreciate your thoughtful comments. The classification of stocking rates was based on local habitat productivity, and the method of GI calculation was based on the number of wether sheep allocated to a specific GI treatment divided by the combined area of 1.5 ha (i.e., 2.7 sheep ha-1= 4 sheep/1.5 ha) for the three replicates in each grazing season (Chen et al. 2010).

Page5 L17-18 Diurnal and seasonal variations of soil respiration were different. Did the authors consider them during the measurement period?

Response from authors: We appreciate the reviewer's suggestion. We considered the diurnal and seasonal variations difference of soil respiration before conduct the experiment. We conducted 22 hour (between 6:00am and 10:00pm, at 2 hour intervals) measurements each day to examine the diurnal soil respiration. The early stage of herbage growth begins in mid May; aboveground grassland biomass peaked in mid September; grassland dormancy occurs by mid December. For these reasons, Rs was measured on six fine days in mid May, September, and December for seasonal variations of soil respiration. Please see L150-L153.

Page7 L15-17 The author can analyze the daily changes of soil respiration of winter grazing and summer grazing under the same grazing gradient to determine which grazing intensity and grazing mode has a great influence.

Response from authors: We appreciate the reviewer's comments that could help us improve our manuscript. As reviewer suggested, we analyze the daily changes of soil respiration of winter grazing and summer grazing under the same grazing gradient. We found grazing intensity with 0 sheep ha-1 has a great influence on soil respiration in warm grazing plots; grazing intensity with 2.7 sheep ha-1 and 5.3 sheep ha-1 has a strong impact on soil respiration in cold grazing plots. We added the analysis in the revised manuscript in L 187-L189, L223-L230 and L667-L669.

Page8 L5 It is suggested that divide this part into two. The first part is to analyze the change of SMBC and SMBN under different grazing intensity and grazing mode, and the second part is to analyze the interaction between abiotic and biotic factors and structural equation model.

Response from authors: Thanks for these constructive suggestions. We added new subtitles which were "3.3 Effects of grazing management on soil temperature, soil moisture, aboveground biomass, belowground biomass" "3.4 Effect of grazing management on temperature sensitivity of soil respiration" "3.5 Structural equation models" in the revised manuscript. Please see L265ïijŇL281 and L 288.

Page9 L27-28 The author has discussed enough. The small suggestion is that soil respiration is closely related to the change of A, and the change of the relationship between the two under different grazing intensity and grazing mode can be considered.

Response from authors: We are grateful for the reviewer's suggestion. We have discussed the annual soil respiration as follows: In addition, the effect of soil temperature on Rs can be explained by the distribution of seasonal precipitation and interannual precipitation. This study found a significant correlation between monthly precipitation and Rs; the precipitation of the semi-arid grassland peaked in September 2011, strongly influencing soil respiration. The significant interannual variations in Rs might be mainly caused by seasonal precipitation fluctuations. Please see L329 – L333.

Figures Fig. 2 The significance test results are not prominent, suggesting recommend the results which is marked significantly and omit unsignificantly.

Response from authors: We greatly appreciate your constructive suggestions. We added the significance test results in figure 2. Please see L 722.

Fig. 5-6 "(d) 5-10 cm in warm season grazing plots" should change to "(d) 5-10 cm in cold season grazing plots" in both Fig. 5 and Fig. 6.

Response from authors:Thanks. Changes were made following your advices in the revised version. Please see L 752-L755 and L761-L764.

Reference

Chen, X. J., Hou, F. J., Matthew, C. & He, X. Z.: Stocking rate effects on metabolizable energy intake and grazing behaviour of Tan sheep in steppe grassland on the Loess Plateau of Northwest China, J. Agr. Sci., 148, 709-721, https://doi.org/10.1017/S0021859610000511, 2010

Please also note the supplement to this comment:
https://www.biogeosciences-discuss.net/bg-2018-531/bg-2018-531-AC3-supplement.pdf

**Supplement:**

**Response of soil respiration and soil microbial biomass carbon and nitrogen to grazing management in semi-arid grassland**

Zhen Wang[1,2], Xiuli Wan[1], Mei Tian[1], Xiaoyan Wang[1], Junbo Chen[1], Xianjiang Chen[1], Shenghua Chang[1], Fujiang Hou[1,2]

[1]State Key Laboratory of Grassland Agro-Ecosystems; College of Pastoral Agriculture Science and Technology, Lanzhou University, Lanzhou 730020, Gansu, China

[2]Key Laboratory of Grassland Agro-Ecosystem, Ministry of Agriculture; Key Laboratory of Grassland Livestock Industry Innovation, Ministry of Agriculture; College of Pastoral Agriculture Science and Technology, Lanzhou University, Lanzhou 730020, Gansu, China

*Correspondence to*: Fujiang Hou. (e-mail: cyhoufj@lzu.edu.cn)

**Abstract**

Grazing management affects grassland carbon dynamics and soil microbial biomass, yet how grazing management, including grazing intensity (GI) and grazing regime (GP), affects soil respiration (Rs) and soil microbial biomass carbon (SMBC) and nitrogen (SMBN) is not fully understood. To determine how GI (0, 2.7, 5.3, and 8.7 sheep ha$^{-1}$) and GP (warm-season grazing, WG; cold-season grazing, CG) affect Rs, SMBC, and SMBN, an experiment was conducted in a semi-arid grassland that had been rotationally grazed for the previous 9 years. Results suggest that diurnal Rs in WG significantly decreased as stocking rate increased; however, in CG, diurnal Rs was significantly higher at the GIs of 2.7 and 5.3 sheep ha$^{-1}$ than at 0 and 8.7 sheep ha$^{-1}$. Although grazing (at the GIs of 2.7, 5.3, and 8.7 sheep ha$^{-1}$) led to increased Rs in 2010 and decreased Rs in 2011, when compared with Rs in the non-grazing period (0 sheep ha$^{-1}$), the negative indirect effect of GI on Rs offset its positive indirect effect on Rs over the whole experimental period. GP affected Rs both directly and indirectly through the positive effect it had on soil moisture, soil temperature, and aboveground biomass. Compared with WG, CG significantly stimulated an increase by 22% in annual Rs. A significant difference in the soil temperature sensitivity ($Q_{10}$)

values of Rs was observed at the four stocking rates for both WG and CG, although the $Q_{10}$ of WG was significantly higher. Interactions between GI and GP had a significant effect on SMBC and SMBN, but GI alone did not affect SMBN. Regarding GP, compared with WG, CG caused a significant decrease of 11% in the mean concentration of SMBN. The monthly precipitation was significantly positively correlated with Rs, soil temperature, soil moisture, and SMBN, but was significantly negatively correlated with SMBC. The field experimental results indicated that the effects of grazing management on Rs processes in the grazing system mainly depend on GP, and the effects of grazing management on SMBC and SMBN mainly depend on the interactions between GI and GP. The results suggest that (1) In a long-term grazing grassland ecosystem, more attention should be paid the role of GP while determining the response of Rs to grazing management, and GP should be considered as an important factor in future evaluation models for studying the response of soil carbon dynamics to climate change; (2) the coupling of GI and GP should be taken into account in future studies on nutrient turnover in the soils of semi-arid grassland ecosystems.

**1. Introduction**

Grasslands cover about 41% of Earth's terrestrial surface, and support domestic livestock grazing in extensive agricultural grazing systems (Morgan et al. 2007). Owing to their large area and excellent ability to sequester and store carbon, grasslands can provide important ecosystem services (Zhao et al. 2017). Grasslands in China constitute 6–8% of global grasslands and significantly contribute to global carbon storage, thereby significantly affecting global carbon cycles (Ni 2002). Soil respiration (Rs) is the second largest carbon flux between the atmosphere and terrestrial biomes (Cox et al. 2000; Wan et al. 2007) and includes microbial and root respiration (Davidson and Janssens 2006; Jia et al. 2007; Ru et al. 2017). Numerous previous studies have indicated that biological and environmental variables, such as precipitation (Xu et al. 2016; Hawkes et al. 2017), soil temperature (Thomey et al. 2011), soil water content (Hawkes et al. 2017), climate, and microbial community composition (Monson et al. 2006), are major factors determining soil and ecosystem respiration in grasslands. In addition, anthropogenic factors, such as grazing, are strongly modifying biogeochemical cycles (Wang and Fang 2009). For example, the structure or species composition of plant communities, soil microclimates, soil chemical and physical properties, and global climate, are all affected by human activities, which in turn might affect Rs rates (Raich and Schlesinger 1992).

Although a growing number of studies have shown that the health of grassland ecosystems strongly depends on grassland management strategies, such as grazing and grazing exclusion (Chen et al. 2015;

Deng et al. 2017), the influence of grazing management on Rs is not well understood. This limits our understanding of how Rs responds to grazing management and our ability to predict carbon dynamic responses under continued climate change. Grasslands are widely distributed on the Loess Plateau, accounting for approximately 40% of the total area (Wang et al. 2017), and are subject to continuous and widespread stress. Most of the grasslands on the Loess Plateau have degenerated owing to their over-grazing and poor management (Fu et al. 2000). Efficient grassland management strategies have been considered to be an important way to promote soil carbon storage and to recover degraded grasslands (Conant et al. 2001; Ingram et al. 2008; Wang et al. 2011). Sustainable grazing intensity (GI)

and a seasonal grazing regime (GP) with periodic resting are widely used grassland management practices (Cui et al. 2014; Chen et al. 2016; Wang et al. 2017, Wu et al. 2017). Grazing-induced soil respiration has been well studied; however, how the effects of different GIs on soil respiration vary through time remains unclear. For example, previous studies have reported that grazing had a positive effect on Rs (Cao et al. 2004), had no effect on Rs (Jia et al. 2007; Wang et al. 2007; Cui et al. 2014), and had a negative effect on Rs (Chen et al. 2016). Furthermore, in different types of grassland, Chen et al. (2015) found that warm-season grazing (WG) decreased Rs, while Wang et al. (2017) reported that cold-season grazing (CG) significantly increased Rs. These inconsistent results could be attributed to the complex processes induced by GP. Therefore, limited information is available on the effects of GP, especially seasonal grazing, on Rs under different levels of GI. Quantifying the effects of GI and grazing season on Rs is critical to accurately estimate the carbon balance of grassland ecosystems and to better understand how global changes affect grazing management practices on the Loess Plateau.

Soil microorganisms are an important component of terrestrial ecosystems and play important roles in global nutrient cycling and organic matter decomposition (Wang et al. 2013). In this context, soil microbial biomass adjusts the balance between the release of carbon during Rs and its sequestration as soil microbial biomass carbon (SMBC) (Lange et al. 2015; Thakur et al. 2015). Thus, microbial biomass is used to assess soil quality. Soil microbial biomass nitrogen (SMBN) is highly labile, and the nitrogen pool in soils is a key regulator of C sequestration (Deng et al. 2017). Elevated soil $CO_2$

reduces available soil nitrogen, imposing nitrogen constraints on microbes and reducing microbial respiration per unit biomass (Hu et al. 2001). Reports on the effects of grazing on SMBC and SMBN in natural grasslands have been inconsistent (Fu et al. 2012; Liu et al. 2012; Lange et al. 2015). Ambient environmental factors (precipitation, air temperature, soil moisture, and soil temperature), livestock type, and grazing management have been shown to affect belowground carbon and nitrogen dynamics (Zhou et al. 2017). Gallardo and Schlesinger (1992) found a succession in the control of microbial biomass from nitrogen to carbon when the ratio of carbon to nitrogen decreased. Fu et al. (2012) found that grazing significantly decreased SMBC and SMBN in an alpine meadow, while Wang et al. (2006) and de Faccio Carvalho et al. (2010) found that cattle grazing increased SMBC. Limited evaluations of soil microbial biomass in semi-arid grasslands and contradictory results regarding the effects of grazing underscore the need for additional research, especially regarding GI (Mahecha et al. 2010) and GP. Therefore, studies on the links between soil microbial biomass and environmental parameters provide a better understanding of the factors that control nutrient cycling in grassland ecosystems.

Given this context, a 2-year study based on a long-term rotational grazing experiment was conducted. The rotational grazing experiment began in 2001, and the Rs and soil microbial C and N were measured from 2010–2011. We hypothesized that the effects of GI on Rs could be offset by GP-induced direct and indirect biotic (above/below-ground biomass, SMBC, SMBN) and abiotic variables (soil temperature, soil moisture) in long-term grazing grassland ecosystems. In the present study, we investigated the following: 1) the effect of long-term rotational grazing on Rs under different GIs and two GPs (warm-season grazing, WG; cold-season grazing, CG); 2) the effect of long-term rotational grazing on SMBC and SMBN under different GIs and GPs; 3) the mechanisms driving the responses of Rs, and soil microbial C and N to the different grazing management practices (GI and GP).

**2. Materials and methods**

The study site was conducted in the core of the Loess Plateau at Huanxian Grassland Ecosystem Trial Station in eastern Gansu Province, northwest China (37.14 °N, 106.84 °E; 1650 m a.s.l.). The average annual air temperature of the study site was 7.1 °C, with maximum temperature occurring in July and minimum in January; the annual mean air temperatures were 8.7 °C and 7.9 °C in 2010 and 2011, respectively (Fig. 2); the average annual rainfall was 360 mm, with >70% of rainfall occurring from mid-June to September (Fig. 2), and the mean monthly precipitations were 23.2 mm and 27.8 mm in and 2011, respectively (Fig. 2), and the precipitation in September 2011 was 104.4 mm, which far exceeded the precipitation in other months (mean monthly precipitation: 27.8 mm). The average annual potential evaporation was 1993 mm. The spring and autumn seasons at the study site were typically short; summer was hot and humid and occurred when most rainfall occurred; winter was long, cold, and dry; soil was classified as sandy, free-draining loess, and the rangeland was a typical temperate steppe (Hou et al. 2002). Dominant grassland species were *Stipa bungeana*, *Lespedeza davurica*, *Pennisetum flaccidum*, *Artemisia capillaris*, and *Setaria viridis* (Hou et al. 2002).

**2.2 Experimental design**

2.2 Experimental design

Two sites with similar topography, vegetation, and cover were selected for separate warm-season (summer) and cold-season (winter) grazing trial plots. Each site was divided into 12 enclosed 0.5-ha replicate plots (i.e., three replicates for each of the four stocking rates); and stocked with 0, 4, 8, and 13 wether sheep of similar liveweight, representing stocking rates of 0, 2.7, 5.3, and 8.7 sheep ha$^{-1}$, respectively (Fig. 1). The classification of stocking rates was based on local habitat productivity, and the method of GI calculation was based on the number of wether sheep allocated to a specific GI treatment divided by the combined area of 1.5 ha (i.e., 2.7 sheep ha$^{-1}$= 4 sheep/1.5 ha) for the three replicates in each grazing season (Chen et al. 2010). The specific methods used for rotational grazing were: In each grazing season, the wethers were allocated to three replicates for each of the four stocking rates, and rotationally grazed between each replicate plot allocated to that stocking rate. The WG plots were rotationally grazed from June to September (90 days), with a rotation cycle length of 30 days (10 days grazing and 20 days rest) and three rotations. The CG plots were grazed from mid-November to late December (48 days), with a rotation cycle length of 24 days (eight days grazing and 16 days rest) and two rotations. The rotational grazing system field experiment began in 2001. The Rs measurements were conducted in both WG and CG plots in 2010, after the trial site had been rotationally grazed for the previous 9 years.

**2.3 Rs measurement**

For the field measurement of Rs, soil $CO_2$ efflux chambers (six PVC collars permanently placed in each plot) were attached to each of the two gas analyzers (LI-COR 8150, Lincoln, NE, USA), which were rotated in a 2-h cycle around all plots in each GP group (WG and CG). The diurnal Rs flux in each plot was measured at 2h intervals between 6:00 am and 10:00 pm, on six fine days in mid-May (early growth stage of herbage), September (peak aboveground biomass), and December (dormancy)

only in 2010 and 2011. Sampling sites were randomly selected at 30–40-m intervals along three 50-m long transects per plot, with two PVC collars (11 cm diameter $\times$ 5 cm height) per transect (i.e., six collars per plot). In preparation for Rs measurement, all aboveground vegetation in each PVC collar was clipped to ensure only Rs was measured. The measurement by gas analyzer took approximately 1.5

min to complete per chamber, connected to six soil efflux collars per plot for a total of 9 min.

**2.4 Soil temperature and soil moisture**

Soil temperature at 10 cm depth was measured in real time using a thermocouple probe attached to every soil efflux collar during each Rs measurement. The soil moisture samples for gravimetric analysis were taken from the top 10 cm, close to every soil efflux collar, once daily in mid-morning (9:00–11:00 h) on the Rs sampling days, and were then oven dried at 105 ℃ for 48 h.

**2.5 Soil sampling and biomass measurements**

Soil samples were collected from random locations adjacent to each Rs measurement site at 5 cm and

10 cm depths during the Rs determination period. After soil samples were coarsely sieved (4.75 mm) to remove rocks and large roots, they were sealed in plastic bags and immediately transported to the laboratory for analysis. The aboveground biomass was estimated by cutting all vegetation in 1 m $\times$ 1 m quadrats (six per plot) after the plots were grazed during the second rotation in early September of 2010

and 2011, when the aboveground biomass peaked. The samples were oven dried at 65 ℃ to a constant weight. Once the aboveground biomass and litter were harvested, soil cores (10 cm depth, 10.0 cm diameter) to a depth of 1 m were collected using a soil auger to calculate the belowground biomass in each quadrat, using the method described by Chen et al. (2015).

**2.6 Soil microbial carbon and nitrogen**

SMBC and SMBN were determined using a chloroform fumigation-extraction procedure and were calculated using the difference in dissolved organic carbon and dissolved organic nitrogen between the fumigated and non-fumigated soil subsamples (Brookes et al. 1985; Vance et al. 1987). Briefly, 10 g soil samples were fumigated with chloroform for 24 h in a vacuum desiccator, and other 10 g samples served as non-fumigated controls. Carbon and nitrogen were extracted with 50 ml of 0.5 M $K_2SO_4$ for

30 min from fumigated and non-fumigated samples, and the extracts were filtered and frozen at −20 ℃

before analysis with a Total Dissolved Organic Carbon and Nitrogen Analyzer-multi NC 2100S

(Analytik Jena AG, Jena, Germany).

**2.7 Statistical analysis**

One-way analyses of variances (ANOVAs) followed by least significant difference (LSD) tests were performed to examine the effects of GI (0, 2.7, 5.3, and 8.7 sheep $ha^{-1}$) and GP (WG and CG) on diurnal fluctuations of Rs. One-way ANOVAs were also used to examine the effect of GI and GP on aboveground biomass, belowground biomass, soil temperature, and soil moisture. The repeated-measures ANOVAs were performed to examine the effect of GI and GP on seasonal variations in Rs ($\mu$mol $CO_2$ $m^{-2}$ $s^{-1}$) and soil microbial carbon (g $kg^{-1}$) and nitrogen (g $kg^{-1}$) in the May,

September, and December of 2010 and 2011 (sampling times). A Pearson correlation analysis was used to test the associations between monthly precipitation and Rs, SMBC, SMBN, soil moisture, soil temperature, and the responses of those variables to monthly precipitation (two-tailed test). Significant differences for all statistical tests were evaluated at the level of $P \leq 0.05$. To investigate the temperature sensitivity of Rs, regression analyses were conducted using Rs = $ae^{bT}$, where Rs is soil respiration, T is soil temperature, coefficient a is the intercept of soil respiration at 0 ℃, and coefficient b represents the temperature sensitivity of Rs that was used to calculate the respiration quotient $Q_{10} = e^{10b}$ (Luo et al.

2001). For means, a two-sample t-test was used to determine the significance of difference between the

$Q_{10}$ values under different GPs. Unless specified, the significance level was set at $P < 0.05$, and uncertainty ($\pm$) always referred to a 95% confidence level. All statistical analyses were conducted using

SPSS 17.0 (SPSS Inc., Chicago, IL, USA).

Structural equation modeling (SEM) was used to evaluate the pathways through which GP and GI

affect Rs both directly and indirectly via biotic and abiotic factors. This was carried out according to the priori conceptual model to include all possible pathways (Supplementary Fig. 1), including (1)

direct and indirect pathways of GP and GI influence on aboveground biomass, belowground biomass, soil temperature, and soil moisture; (2) direct and indirect pathways of GP and GI influence on soil microbial carbon and nitrogen; and (3) direct and indirect pathways of GP or GI influence on Rs via biotic or abiotic factors. Before constructing the SEM models, a correlation matrix was derived for all variables using least-squares. To differentiate the effects of grazing management on Rs, grazing management was divided into two sections. The first SEM was based on bivariate regressions to focus on the direct and indirect effects of GI on Rs, and included aboveground biomass, belowground biomass, soil moisture, soil temperature, SMBC, and SMBN. The second SEM focused on both direct and indirect effects of GP on Rs. After first considering a full model that included all possible pathways, non-significant pathways were sequentially eliminated, until arriving at the final model. The $\chi^2$ test, Akaike information criteria, and root mean square error of approximation were used to evaluate the fit of model. The SEM analyses were conducted using AMOS 17.0 (SPSS Inc., Chicago, IL, USA).

**3. Results**

**3.1 Diurnal and seasonal dynamics of Rs rate under four different GIs and two GPs**

Daily maximum, minimum, and total Rs rates under the four GIs in the two grazing seasons are shown in Supplementary Tables 1 and 2, respectively. GI significantly influenced the diurnal variation of Rs in the WG (Table 1, $P <0.001$) and CG plots (Table 1, $P <0.001$). In a WG plot, the diurnal Rs significantly decreased as the stocking rate increased (Table 1, $P <0.001$), but in a CG plot, the diurnal Rs was significantly higher at the GIs of 2.7 and 5.3 sheep ha$^{-1}$ than at 0 and 8.7 sheep ha$^{-1}$ (Table 1, $P <0.001$). The diurnal Rs was not significantly different between the WG and CG plots in the non-grazing period (Table 1, 0 sheep ha$^{-1}$, $P = 0.964$); however, the Rs of a WG plot was significantly lower than that for a CG plot in the grazing period (Table 1, 2.7, 5.3, and 8.7 sheep ha$^{-1}$, $P <0.001$), indicating that CG increases diurnal soil Rs.

From 2010 to 2011, Rs showed obvious seasonal and interannual changes in both WG and CG plots (Fig. 3, Table 2). Compared with the non-grazed plots (0 sheep ha$^{-1}$), GI significantly increased the Rs between sampling seasons by approximately 22% in 2010 (Table 2, $P = 0.007$), but decreased it by approximately 16% in 2011 (Table 2, $P = 0.011$). The interactions between the GIs and sampling seasons showed no effect on Rs in 2010 and 2011. The GIs (Table 2, $P = 0.826$) and the interactions between the GIs and years (Table 2, $P = 0.070$) had no significant effect on Rs over the whole experiment period. A significant difference in Rs was observed between the WG and CG plots in 2010 (Fig. 3, Table 2, $P <0.001$), but not in 2011 (Table 2, $P <0.964$). In both years, GP and year, as well as their interactions, had a significant effect on Rs ($P <0.001$), and Rs was approximately 22% higher in the CG plots than in the WG plots ($P$ <0.001) and was significantly and positively correlated with monthly precipitation (Table 3, $P$ <0.001).

**3.2 SMBC and SMBN**

SMBC and SMBN at 5 and 10 cm soil depths were characterized by pronounced temporal dynamics between and within sampling seasons over the investigated time across 2010 and 2011 (Figs 4 and 5). Compared with WG, CG significantly increased SMBC by 6% in 2010 (Table 2, $P$ <0.001), but decreased it by 19% in 2011 (Table 2, $P$ <0.001). GI affected SMBC neither in 2010 (Table 2, $P$ = 0.129), nor in 2011 (Table 2, $P$ = 0.208). SMBC was significantly affected by the interactions between GPs and GIs in 2011 (Table 2, $P$ <0.001) and by the GPs and sampling seasons and their interactions in both 2010 (Table 2, $P$ <0.05) and 2011 (Table 2, $P$ <0.05). In both years, GP and GI alone did not significantly affect SMBC (Table 2, $P$ >0.05), but their interactions had a significant effect on SMBC (Table 2, $P$ <0.001). Moreover, SMBC was significantly affected by the interaction between the year and GP (Table 2, $P$ <0.001), but not by the interaction between the year and GI (Table 2, $P$ = 0.224). When compared with 2010, the mean concentrations of SMBC decreased by 58% and 69% in the WG and CG plots, respectively, in 2011 (Fig. 4). SMBC was significantly negatively correlated with monthly precipitation (Table 3, P <0.001). SMBN was significantly affected by GP (Table 2, $P$ <0.05), sampling season (Table 2, $P$ <0.001), and the interactions between GI and GP (Table 2, $P$ <0.05) in 2010 and 2011. GI alone significantly affected SMBN (Table 2, $P$ = 0.042) in 2011, but not in 2010 (Table 2, $P$ = 0.601). In both years, a significant difference in SMBN was observed between the WG and CG plots ($P$ <0.001); the interaction between GP and GI had a significant effect on SMBN (Table 2, $P$ = 0.007). Relative to WG, CG significantly decreased the mean concentration of SMBN by 11% over the entire study period (Table 3, $P$ <0.001). SMBN was significantly positively correlated with monthly precipitation (Table 3, $P$ <0.001).

**3.3 Effects of grazing management on soil temperature, soil moisture, aboveground biomass, and belowground biomass.**

The diurnal variation in soil temperature, measured at 2-h intervals at depths of 0–10 cm, changed significantly over time (Fig. S1, $P$ <0.05). No significant differences in soil temperature were observed in both CG and WG plots under different stocking rates (2.7, 5.3, and 8.7 sheep ha$^{-1}$) ($P$ = 0.63). Soil temperature in the CG plots was 3.1 ℃ higher ($P$ = 0.02) than that in the WG plots (Fig. S2). The soil temperature varied significantly ($P$ <0.001) by year, with the highest values observed in 2011 and the lowest in 2010 (Fig. S2). The annual mean soil moisture in 2011 was 7.9% higher ($P$ <0.001) than that in 2010 (Fig. S3). GI did not affect soil moisture ($P$ = 0.87); however, the CG plots increased soil moisture by 13.2% compared with that in the WG plots ($P$ = 0.005). A significant posititive correlation was observed between monthly precipitation and soil temperature and soil moisture (Table 3, $P$ <0.001).

The GI significantly decreased the aboveground (Fig. S4, $P$ <0.001) and belowground biomass (Fig. S4,

$P$ = 0.001). GP significantly affected the aboveground biomass ($P$ <0.001), but not the belowground biomass ($P$ = 0.071). The aboveground biomass was higher in the CG plots than in the WG plots (by

16%).

**3.4 Effect of grazing management on temperature sensitivity of soil respiration ($Q_{10}$)**

During the whole experimental period, a significant exponential positive correlation between the rates of Rs change and soil temperature was observed in both WG plots and CG plots, indicating that Rs is strongly influenced by soil temperature ($P$ <0.001, Table 3). A significant difference in the $Q_{10}$ values of Rs was observed among the GIs in both WG and CG plots ($P$ <0.001, Table 3). With regard to the

GPs, the $Q_{10}$ values of the WG plots were significantly higher than those of the CG plots (Table 3).

**3.5 Structural equation models**

The structural equation models showed that Rs was indirectly explained by GI, which explained 57%

of the total variation in Rs in the grazing ecosystem (Figs 6 and 7). GI decreased Rs via its negative effect on aboveground biomass, whereas it increased Rs via its positive effect on soil microbial biomass (Fig. 6). These negative and positive indirect effects of GI on Rs offset the direct effect of GI

on Rs (Fig. 6). GP explained 57% of the total variation in Rs in the grazing ecosystem (Fig. 7). We not only observed a direct positive effect of GP on Rs, but also an indirect effect of GP on Rs via its positive effects on soil moisture, soil temperature, and aboveground biomass. In addition, the negative effect of GP on Rs via its negative effect on SMBC and SMBN weakened the positive influence of GP

on Rs.

**4.  Discussion**

**4.1 Effect of grazing management on Rs**

The diurnal changes in Rs under different GPs (warm and cold season) with four GIs (0, 2.7, 5.3, and 8.7 sheep ha$^{-1}$) observed in our study were within the range of the daily Rs reported by several previous studies (Zhang et al. 2014; Wang et al. 2015; Rong et al. 2017). We found that GI significantly affected the diurnal changes of Rs in both WG ($P$ <0.001) and CG plots ($P$ <0.001), but the response was different. Since the diurnal changes in Rs followed a unimodal pattern through time, consistent with soil temperature, this could be due to the differences in sensitivity to temperature in grazing seasons or could be due to the spatial heterogeneity of Rs (Wang et al. 2013; Wang et al. 2017). Our study revealed that CG enhances the diurnal Rs rate. These results are consistent with those of a previous study that performed a meta-analysis of Tibetan grasslands (Wang et al. 2017). The GI had a significant effect on seasonal changes in Rs in both 2010 ($P$ = 0.007) and 2011 ($P$ = 0.011); however, the interaction between season and GI had no significant effect on Rs in 2010 ($P$ = 0.253) and 2011 ($P$ = 0.153). It was also found that both GI ($P$ = 0.826) and the interaction between GI and year ($P$ = 0.070) had no effect on Rs throughout the whole experimental period; this could be explained by both biotic and abiotic pathways (Fig. 6) in the following two ways: 1) The negative impact of GI on Rs was achieved indirectly by directly negatively affecting the aboveground biomass or by indirectly negatively affecting the microbial biomass; 2) At the same time, GI had a positive and indirect effect on Rs, which was achieved by directly negatively affecting the belowground biomass or by directly affecting soil moisture. Consequently, these two different types of pathways offset each other within a certain range.

The results of this study indicate that GP ($P$ <0.001), year ($P$ <0.001), and their interaction ($P$ <0.001) have strong effects on Rs. Compared to WG, CG promoted soil respiration by 22%. These results were supported by the SEM analyses performed in this and previous studies (Jiang et al. 2010; Wang et al. 2013; Chen et al. 2015; Xu et al. 2016). The response of Rs to GP can be explained by the following three mechanisms (Fig. 7): (1) The CG increased soil temperature by 3.1 ℃ and increased soil moisture by 13.2%, and both soil temperature and soil moisture had direct positive effects on Rs. Soil temperature and water availability effect Rs by altering the activity of plant roots and soil microbes, and they also indirectly affect soil respiration by altering plant growth and substrate supply (Wan et al.

2007). In addition, the effect of soil temperature on Rs can be explained by the distribution of seasonal precipitation and interannual precipitation (Ru et al. 2017). This study found a significant correlation between monthly precipitation and Rs (P <0.001); the precipitation of the semi-arid grassland peaked in September 2011, strongly influencing soil respiration. The significant interannual variations in Rs might be mainly caused by seasonal precipitation fluctuations; this could be proved by setting up a controlled field experiment. (2) Compared with WG, CG significantly increased the aboveground biomass ($P$ <0.001), which directly affected Rs. This might be due to the increased sensitivity of the aboveground biomass to low temperatures (Abdalla et al. 2010). On one hand, CG reduces litter and increases sunshine exposure, which is beneficial for plant growth in the subsequent year (Coughenour 1991; Altesor et al. 2005; Wang et al. 2017). On the other hand, grazing leads to a warm and dry microclimate by removing aboveground plants and compacting the soil (Rong et al. 2017). (3) GP has a negative effect on Rs through its negative effects on soil microbial biomass and microbial nitrogen. Since Rs is a process of converting organic carbon to inorganic carbon, the rate of Rs is ultimately controlled by the supply of carbon substrates (Xu et al. 2016; Bagchi et al. 2017). Soil microbial communities exhibit high substrate utilization rates at low temperatures (Monson et al. 2006); thus, a decrease in microbial biomass will reduce soil carbon emissions (Allison et al. 2010). Previous studies have reported that SMBC primarily determines microbial respiration (Zhang et al. 2014). Significant changes in SMBC and SMBN have significant effects on $CO_2$ emissions under different land-use patterns (Iqbal et al. 2010).

Previous studies have been conducted on the temperature sensitivity of Rs ($Q_{10}$), both globally and in different ecosystems of China (Luo et al. 2001; Curiel et al. 2004; Davidson et al. 2006; Chen et al. 2015; Chen et al. 2016). In our study, under different grazing rates, the $Q_{10}$ values ranged between 1.31 and 1.57 for the WG plots and between 1.21 and 1.36 for the CG plots, agreeing with the results of Chen et al. (2015). The $Q_{10}$ values of the WG plots were higher than those of the CG plots. This might be due to soil water freezing at low temperatures, which inhibits soil microbial activity, and thus reduces $Q_{10}$ in the CG seasons (Mahecha et al. 2010; Chen et al. 2016). The $Q_{10}$ values for different ecosystems on a global scale are different, but the median is 1.4 (Mahecha et al. 2010). Our results are similar to this median, indicating that it is necessary to consider the impact of Rs on climate warming under different GIs and different grazing systems in order to accurately assess the carbon cycle of semi-arid grassland ecosystems. Overall, our results indicate that the mechanisms underlying the effects of grazing management on Rs mainly depend on GP, but not on GI. This indicates that the effects of GP, especially seasonal and long-term grazing, should be considered in future manipulation experiments and carbon models to accurately simulate soil carbon dynamics in semi-arid grassland ecosystems.

**4.2 Effects of grazing management on SMBC and SMBN**

The results of our study revealed that SMBC was higher at the beginning (May) and in the middle (September) of the growing season than in the dormant period (December) (Fig. 4). These results are consistent with the findings of previous studies conducted in the grassland ecosystems of the trans-Himalaya (Bagchi et al. 2017) and the Tibetan Plateau (Fu et al. 2012). We found that SMBN in the WG plots was significantly higher than that in the CG plots ($P$ <0.001). The reason for this might be that the response of SMBN is more sensitive to grazing than that of SMBC (Fu et al. 2012). CG increased the soil temperature, which increased microbial biomass (Lu et al. 2013; Wang et al. 2017). Our results show that GI did not have an effect on SMBC ($P$ >0.05) and SMBN ($P$ >0.05), but the interactions between GI and GP significantly affected SMBC ($P$ <0.001) and SMBN ($P$ <0.007). These results indicate that the response of SMBC is coupled with GI and GP. On one hand, high GI increases bulk density and urine input and decreases soil porosity and aggregation, affecting microorganism metabolism (Prieto et al. 2011; Liu et al. 2012,). On the other hand, GI leads to a variation in SMBC as a direct result of the soil water content from higher precipitation and temperature.

According to our SEM analysis, there are two pathways to explain the effects of GI (Fig. 6) and GP (Fig. 7) on SMBC and SMBN. (1) The effect of GI on SMBC and SMBN occurs mainly via its adverse effects on aboveground biomass, which directly stimulates SMBC. Recent studies have shown that the effects of grazing on soil microbial community size are largely dependent on GI via biotic factors (Zhao et al. 2017); grazing decreases the aboveground and belowground biomass (Koerner and Collins 2014). Our results support the theory that grazing management could change soil microbial activities by regulating the aboveground and belowground biomass, which in turn changes the microbial biomass in the soil (Stark et al. 2015; Xu et al. 2017). (2) GP positively affects soil temperature and soil moisture, both of which stimulate SMBC and SMBN. In our study, compared with WG, CG significantly increased soil moisture ($P$ = 0.005) and soil temperature ($P$ = 0.020), which might stimulate more efficient enzymes to catalyze the reactions of soil organic matter decomposition (Stark et al. 2015). Moreover, the dissolved organic carbon was metabolized only after rewetting, and the chemical signals released by the roots regulate the microbial communities, some of which have powerful feedbacks in carbon cycling (Schimel et al. 2013). Monthly precipitation events were significantly negatively related to SMBC ($P < 0.001$), but significantly positively related to SMBN in our study ($P < 0.001$). This agrees with the previous studies that found that the precipitation events stimulating microbial activity might shift the C-balance of grassland ecosystems (Curiel et al. 2007) and that grazing interacts with precipitation to affect the belowground biomass (Koerner and Collins 2014). Overall, our results indicate that the effects of grazing management on SMBC and SMBN mainly depend on the interactions between GI and GP. This suggests that integrated grazing management strategies should be taken into account in future studies on nutrient turnover in the soils of semi-arid grassland ecosystems.

*Author contributions.* Fujiang Hou designed and directed the study, Zhen Wang carried out the data analysis, and wrote the manuscript. Xiuli Wan, Junbo Chen, Mei Tian, Xiayan Wang, Xianjiang Chen, Shenghua Chang collected samples, analyzed the data, and contributed to the final writing of the manuscript.

*Data availability.* from the corresponding author, Fujiang Hou (cyhoufj@lzu.edu.cn), upon request.

*Competing interest.* The authors declare that they have no conflicts of interest.

*Acknowledgements.* We wish to thank for the help in the revision of the manuscript. This work was supported by the Program for Changjiang Scholars and Innovative Research Team in University (IRT17R50), the National Key Basic Research Program of China (2014CB138706), the National Natural Science Foundation of China (31172249), the Strategic Priority Research Program of Chinese Academy of Sciences (XDA2010010203), and the 111 project (B12002).

**Table 1.** Effect of warm-season grazing (WG) and cold-season grazing (CG) on daily soil respiration (Rs) under different grazing intensities (GI). F- and P-values were obtained by the one-way ANOVA analyses. Different letters in the same column indicate statistical significant differences between grazing intensity under certain grazing season (P < 0.05). Data are shown as the mean $\pm$ SE (n=800).

| GI (sheep ha$^{-1}$) | Rs ($\mu$ mol $CO_2$ m$^{-2}$ s$^{-1}$) | | | | |
|---|---|---|---|---|---|
| | WG | CG | df | F-Value | P-Valve |
| 0 | 0.91 ±0.02a | 0.91 ±0.02c | 1.00 | 0.002 | 0.964 |
| 2.7 | 0.81 ±0.02b | 1.30 ±0.02a | 1.00 | 307.444 | **0.001** |
| 5.3 | 0.76 ±0.02c | 1.28 ±0.02a | 1.00 | 380.222 | **0.001** |
| 8.7 | 0.76 ±0.02c | 1.18 ±0.02b | 1.00 | 288.455 | **0.001** |

**Table 2.** Effects of grazing regime (GP), grazing intensities (GI), year, sampling season , and their interactions on the rates of Rs, SMBC, and SMBN. F-and P-values are obtained by the repeated-measures ANOVA analyses. Values in bold indicate significance levels of $P < 0.05$.

| Year | Factors | Rs | | | SMBC | | | SMBN | | |
|---|---|---|---|---|---|---|---|---|---|---|
| | | df | F Value | P | df | F Value | P | df | F Value | P |
| **2010** | GP | 1 | 83.672 | **<0.001** | 1 | 21.270 | **<0.001** | 1 | 5.008 | **0.027** |
| | GI | 3 | 4.234 | **0.007** | 3 | 1.922 | 0.129 | 3 | 0.623 | 0.601 |
| | GI×GP | 3 | 9.427 | **<0.001** | 3 | 44.321 | **<0.001** | 3 | 5.976 | **<0.001** |
| | Season | 2 | 218.389 | **<0.001** | 2 | 68.065 | **<0.001** | 2 | 25.685 | **<0.001** |
| | GP×Season | 2 | 6.186 | **0.003** | 2 | 15.389 | **<0.001** | 2 | 5.976 | **<0.001** |
| | GI×Season | 6 | 1.317 | 0.253 | 6 | 3.828 | **<0.001** | 6 | 4.515 | **0.013** |
| | GI×GP×Season | 6 | 0.867 | 0.521 | 6 | 5.417 | **<0.001** | 6 | 2.192 | **0.047** |
| | | | | | | | | | | |
| **2011** | GP | 1 | 0.002 | 0.964 | 1 | 28.930 | **<0.001** | 1 | 12.725 | **<0.001** |
| | GI | 3 | 3.853 | **0.011** | 3 | 1.534 | 0.208 | 3 | 2.812 | **0.042** |
| | GI×GP | 3 | 0.205 | 0.893 | 3 | 2.186 | 0.092 | 3 | 2.966 | **0.034** |
| | Season | 2 | 333.087 | **<0.001** | 2 | 29.162 | **<0.001** | 2 | 81.204 | **<0.001** |
| | GP×Season | 2 | 6.640 | **0.002** | 2 | 3.353 | **0.038** | 2 | 20.360 | **<0.001** |
| | GI×Season | 6 | 1.593 | 0.153 | 6 | 3.682 | **0.002** | 6 | 9.269 | **<0.001** |
| | GI×GP×Season | 6 | 1.498 | 0.183 | 6 | 2.955 | 0.009 | 1 | 0.000 | 0.526 |
| | | | | | | | | | | |
| **Overall** | GP | 1 | 12.391 | **<0.001** | 1 | 0.197 | 0.658 | 1 | 18.129 | **<0.001** |
| | GI | 3 | 0.299 | 0.826 | 3 | 2.470 | 0.062 | 3 | 1.873 | 0.134 |
| | GI×GP | 3 | 1.395 | 0.244 | 3 | 22.313 | **<0.001** | 3 | 4.154 | **0.007** |
| | Year | 1 | 36.942 | **<0.001** | 1 | 1280.544 | **<0.001** | 1 | 0.995 | 0.319 |
| | GP×Year | 1 | 16.867 | **<0.001** | 1 | 16.170 | **<0.001** | 1 | 3.473 | 0.063 |
| | GI×Year | 3 | 2.375 | 0.070 | 3 | 1.396 | 0.244 | 3 | 2.051 | 0.107 |
| | GI×GP×Year | 3 | 1.867 | 0.135 | 3 | 20.096 | **<0.001** | 3 | 3.195 | **0.024** |

**Table 3.** Pearson correlation between monthly precipitation (P) and soil temperature (ST),soil moisture (SM) soil respiration (Rs), soil microbial biomass carbon (SMBC) and soil microbial biomass nitrogen (SMBN). *** indicate significance level at $P < 0.001$.

| Variables | ST | SM | SR | SMBC | SMBN |
|---|---|---|---|---|---|
| P | 0.298*** | 0.737*** | 0.475*** | -0.162*** | 0.543*** |

**Table 4.** Temperature sensitivity of soil respiration ($Q_{10}$) for the grazing regimes (warm-season grazing, WG; cold-season grazing, CG) with different grazing intensities (GI, 0, 2.7, 5.3, 8.7 sheep ha$^{-1}$). a and b are two coefficients in the regression line Rs = ae$^{bT}$, where Rs is soil respiration and T is soil temperature. r$^2_{adj}$ is the adjustive determinant coefficient, Q$_{10}$ is the temperature quotient (= e10$^b$). Significance levels of *P* < 0.05 indicated in bold.

| GP | GI | a | b | r$^2_{adj}$ | P | Q$_{10}$ |
|---|---|---|---|---|---|---|
| | (sheep ha$^{-1}$) | | | | | |
| WG | 0 | 0.510 | 0.027 | 0.212 | **<0.001** | 1.310 |
| | 2.7 | 0.318 | 0.045 | 0.478 | **<0.001** | 1.568 |
| | 5.3 | 0.313 | 0.043 | 0.458 | **<0.001** | 1.537 |
| | 8.7 | 0.343 | 0.039 | 0.373 | **<0.001** | 1.477 |
| CG | 0 | 0.473 | 0.031 | 0.271 | **<0.001** | 1.363 |
| | 2.7 | 0.689 | 0.019 | 0.155 | **<0.001** | 1.209 |
| | 5.3 | 0.643 | 0.022 | 0.200 | **<0.001** | 1.246 |
| | 8.7 | 0.598 | 0.023 | 0.196 | **<0.001** | 1.259 |

[Figure]

**Figure 1.** Long-term rotaional grazing experimental design at the study site. Green box represent warm season grazing plots; brown boxes represent cold season grazing plots..

[Figure]

**Figure. 2.** Temporal variation in measured values of air temperature, precipitation, at the study site from January 2010 to December 2011. Straight line and line of dashes represents mean annual precipitation represents and mean annual air temperature from 2001 to 2009. The vegetation of the grassland starts to regreen in late April to early May, and starts to wither in late October. Air temperature and precipitation did not change significantly during 2010 and 2011.

[Figure]

**Figure. 3.** Seasonal dynamics of Rs with different grazing intensities (a) in warm season grazing area, (b) in cold season grazing area. Vertical bars indicate mean bars standard errors (n=9).

[Figure]

**Figure. 4.** Seasonal dynamics of Soil microbial biomass carbon (SMBC) (a) 0-5 cm soil layers in warm-season grazing plots; (b) 0-5 cm in cold-season grazing plots; (c) 5-10 cm in warm-season grazing plots; (d) 5-10 cm in cold-season grazing plots. Vertical bars represent the standard error of the measurement mean (n =3) for each observation date.

[Figure]

**Figure. 5.** Seasonal dynamics of soil microbial biomass nitrogen (SMBN) (a) 0-5 cm in warm-season grazing plots; (b) 0-5 cm in cold-season grazing plots; (c) 5-10 cm in warm-season grazing plots; (d) 5-10 cm in cold-season grazing plots. Vertical bars represent the standard error of the measurement mean (n =3) for each observation date.

**Figure. 6.** Structural equation model of grazing intensities (Mahecha et al.) effects on soil respiration (Rs) via direct or indirect effect on aboveground biomass (AM), belowground biomass(BM), soil temperature (ST), SM (soil moisture), SMBC (soil microbial carbon), soil microbial nitrogen (SMBN). Red and black arrows represent significant negative and positive pathways, respectively. Bold numbers indicate the standard path coefficients. Arrow width is proportional to the strength of the relationship. $R^2$ represent the proportion of variance explained for each dependent variable in the model. ***P<0.001, **P<0.01, *<0.05; $\chi$=10.746; P=0.057; root mean square error of approximation (RMSEA) =0.059; P =0.057; Akaike information criteria (AIC) =88.746.

[Figure]

**Figure. 7.** Structural equation model of grazing regime (GP) effects on soil respiration (Rs) via direct or indirect effect on aboveground biomass (AM), soil temperature (ST), SM (soil moisture), SMBC (soil microbial carbon), soil microbial nitrogen (SMBN). The structural equation model considered all plausible pathways through which experimental treatments influence Rs. Red and black arrows represent significant negative and positive pathways, respectively. Bold numbers indicate the standard path coefficients. Arrow width is proportional to the strength of the relationship. $R^2$ represent the proportion of variance explained for each dependent variable in the model. ***$P<0.001$, **$P<0.01$, *$<0.05$; $\chi=2.418$; $P=0.299$; root mean square error of approximation (RMSEA) $=0.025$; $P =0.299$; Akaike information criteria (AIC) $=68.418$.